



# Very short-term forecast of near-coastal flow using scanning lidars

Laura Valldecabres[1], Alfredo Peña[2], Michael Courtney[2], Lueder von Bremen[1], and Martin Kühn[1]

[1]ForWind - University of Oldenburg, Institute of Physics, Küpkersweg 70, 26129 Oldenburg, Germany
[2]DTU Wind Energy, Risø Campus, Technical University of Denmark, Frederiksborvej 399, 4000 Roskilde, Denmark

*Correspondence to:* Laura Valldecabres (laura.valldecabres@forwind.de)

**Abstract.** Wind measurements can reduce the uncertainty in the prediction of wind energy production. Nowadays, commercially available scanning lidars can scan the atmosphere up to several kilometres. Here, we use lidar measurements to forecast near-coastal winds with lead times of five minutes. Using Taylor's frozen turbulence hypothesis together with local topographic corrections, we demonstrate that wind speeds at a downstream position can be forecast by using measurements from a scanning lidar performed upstream in a very short-term horizon. The study covers ten periods characterized by neutral and stable atmospheric conditions. Our methodology shows smaller forecasting errors than those of the persistence method and the ARIMA model. We discuss the applicability of this forecasting technique with regards to the characteristics of the lidar trajectories, the site-specific conditions and the atmospheric stability.

## 1 Introduction

Wind energy is growing worldwide as a major source of green energy. In 2015 Denmark produced a record 42 % of the country's electricity with wind energy (REN21, 2016). As the share of variable energy into the grid grows, more effort is required to increase the flexibility of power systems in such a way that they can guarantee the grid stability (Holttinen et al., 2016) and the reliability of energy supply (Ibanez and Milligan, 2012). For power systems with high penetration of intermittent renewable energy, one of the most important sources of imbalances is wind energy forecast errors (Gonzalez-Aparicio and Zucker, 2015). On short-time scales, Transmission System Operators maintain the balance between electricity production and demand, activating balancing reserves. Due to their high flexibility to respond to short-term changes in power, balancing reserves mostly come from conventional power plants, which reduces the environmental and economic benefits of wind energy. In countries like Belgium, the Netherlands and Germany, the electricity market participants can submit their intraday bids until five minutes before delivery (EPEXSPOT, 2017). With potentially shorter gate closure times wind power suppliers can better match production with demand, thus minimizing the costs arising from the deviation between scheduled wind energy production and real generation (Wang et al., 2016).

In very short-term horizons, i.e. from minutes to one hour, wind forecasts are normally based on statistical models. They are built on relationships developed between historical measurements, assuming that these relationships are also applicable in the future. Examples of statistical methods used to predict wind speed and power can be found in Hill et al. (2012) for the autoregressive (AR) model, Torres et al. (2005) for the autoregressive moving average (ARMA) model and Kavasseri and Seetharaman (2009) for the autoregressive integrated moving average (ARIMA) model. Torres et al. (2005) applied the ARMA





model to predict hourly average wind speeds in five weather stations in Navarre, Spain during different times of the year, with a forecasting horizon from one to ten hours. They showed smaller errors for ARMA models compared to those of the persistence method. The classical persistence model predicts that the future value will be the same as the current value. This is the simplest version of the ARMA model and is often considered as a benchmark for other forecasting techniques (Giebel et al., 2011).

Another forecasting technique is the spatial correlation method, which uses the wind speed at upstream neighbouring points to predict the wind speed at a downstream location. First spatial correlation models were developed by Schlueter et al. (1986). They predicted meteorological events based on cross-correlation curves of wind speeds at two sites using a constant delay method. Alexiadis et al. (1998) tested this method in the Greek islands of Syros and Paros in a time horizon of 10 min to some hours. Although there was a high correlation between the two sites in terms of fluctuations, the errors were higher than those

of the persistence method both in magnitude and phase. They later proposed a spatial correlation predictor method, which uses linear relations to correct magnitude and phase errors.

Recently, various techniques based on artificial neural networks, which are trained with large historic data from the location, were developed (Cadenas and Rivera, 2009; Monfared et al., 2009; Li and Shi, 2010). Damousis et al. (2004) implemented a wind forecasting fuzzy model in which wind data from neighboring meteorological stations at a radius up to 30 km were used

to predict wind speed and power in horizons of 30 min to two hours. The model results showed significant improvement in the forecasting error of wind speed and power compared to those of the persistence model especially, when applied on flat terrain.

Wind forecasting techniques can combine physical and statistical approaches. As an example, Larson and Westrick (2006) used off-site observations at the vicinity of a wind farm in north-eastern Oregon, as input variables in different forecast models such as neural networks and support vector machines. They showed that the integration of real-time off-site observations

significantly improves the forecasting accuracy of those algorithms.

Nowadays, remote sensing systems like lidars are intensively being deployed for wind resource assessment (Wharton et al., 2015), turbine control (Mikkelsen et al., 2013) and turbulence characterization (Peña et al., 2017). Lidars are proven to be relevant for very short-term forecasting (Frehlich, 2013) as the current generation of commercially available units can scan in various atmospheric conditions up to 30 km. As an example, a $4\,\mathrm{m\,s^{-1}}$ wind speed could be observed by a lidar located in a

wind farm 3.6 km upstream, 15 min ahead, thus predicting the start of power generation. Remote sensing systems could also be used to better schedule maintenance of offshore wind farms (Barthelmie et al., 2008), e.g. during periods of low wind speeds.

With our study we want to (i) experimentally investigate how lidar observations can be used to forecast wind speeds in a very short-term horizon assuming Taylor's frozen turbulence hypothesis (Taylor, 1938) and (ii) test if with the use of lidar measurements we can predict wind speeds better than with the benchmarks ARIMA and persistence model. For this, we

use lidar observations up to 6 km in a near-coastal area in Denmark as an input for an advection-based wind speed forecast technique. The observations are characterized by rather high wind speeds, which limit the forecasting horizon to 5 min. The lidar measurements and the data analysis are described in Sect. 2. An insight into the wind conditions is given in Sect. 3. The location of the lidars in the near-coastal area made necessary to consider the topographic local conditions, which are modelled in Sect. 4. Section 5 gives a detailed description of the methodology used to forecast wind speeds using the lidar

measurements. In Sect. 6 results are presented through comparisons between the accuracy of forecasting wind speeds based





on the advection models with persistence and ARIMA models. We discuss the suitability of using long-range lidars for very short-term forecasting and provide main conclusions in Sect. 7.

## 2   Wind data analysis

Our study is based on measurements performed during the Reducing Uncertainty of Near-shore wind resource Estimates
(RUNE) campaign (Simon and Courtney, 2016; Floors et al., 2016). The experiment was conducted at the western coast of Denmark, North of the area of Høvsøre (see Fig. 1). A comprehensive analysis of the wind conditions at Høvsøre during a ten years period from a meteorological mast 1.7 km east of the North Sea (see Fig. 1) is presented in Peña et al. (2016). A pronounced cliff at the coastline (see Fig. 2) is the main feature of the terrain, which is mainly covered with grass and crops.

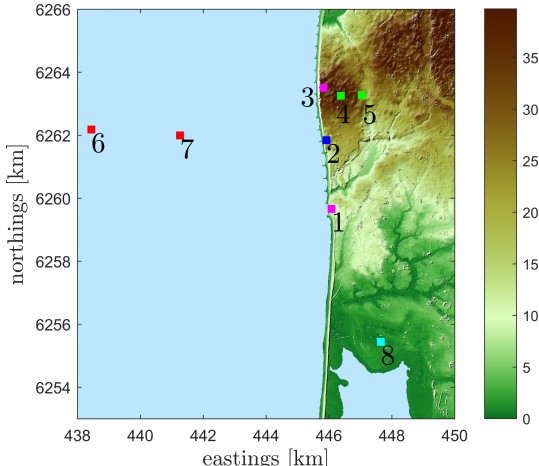

**Figure 1.** Map of the area of the RUNE campaign indicating the positions of the measurement systems (see Table. 1).

During the RUNE campaign, which took place during November 2015 to February 2016, profiling and scanning lidars were
deployed to measure near-coastal wind conditions (see Fig.1). Four short-range (positions 2, 4, 5 and 6 (later 7)), and one long-range (position 2) profiling lidars measured the wind profile. One scanning lidar (position 2) was operated in Plan Position Indicator (PPI) mode, also known as the "sector-scan" scenario. Simultaneously, two more scanning lidars (at positions 1 and 3) were configured in a dual trajectory to match at positions along three horizontal virtual lines. In what follows we will refer to them as the dual-setup. In Fig. 2 the positions of the dual-setup and the PPI are shown. The PPI and the dual-setup trajectories
were designed so that the measurements will intersect at 5000 m offshore at 50, 100 and 150 m AMSL. Further, a directional wave buoy was deployed to measure waves, currents and see surface temperature. Detailed information about the campaign can be found in Floors et al. (2016). For this work we also use data from the sonic and cup anemometers located at the height of 100 m on the Høvsøre meteorological mast. Table 1 summarizes the operational availability of all systems used in this study.



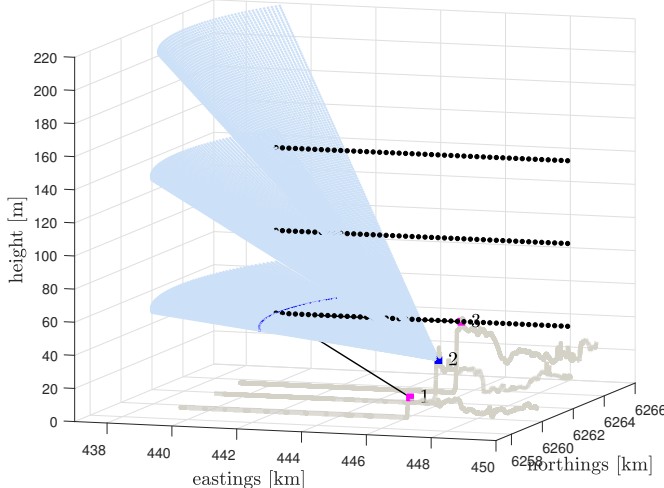

**Figure 2.** Scanning trajectories of the dual-setup (black dots) and the PPI (light blue points) scenarios. The black lines show two laser beams from the dual-setup lidars (magenta boxes) focusing at position 7 (see Fig. 1). The grey lines indicate the terrain height above sea level.

## 2.1 Lidar data processing and filtering

### 2.1.1 Dual-setup measurements

The two lidars measuring in the dual-setup trajectory acquired 45 line-of-sight (LOS) wind velocities (1 s per LOS) per horizontal virtual line, separated by a distance of $\approx 200$ m between points from 4 km onshore to 5 km offshore. Every trajectory,

i.e. three horizontal lines at 50, 100 and 150 m AMSL, took 145 s. In total, every position was swept 4 times every 10 min. Data that did not fulfil a certain distance threshold between the two lidar measurement positions were discarded. Regarding data quality, a carrier-to-noise ratio (CNR) threshold of $-26.50$ dB was set. For every 10-min period and each point, the horizontal wind speed components were reconstructed as described in Simon and Courtney (2016). Due to the low availability in the reconstruction at positions further away from the coast (Floors et al., 2016), we only consider data up to 2950 m. Observations

close to the lidar systems (range $< 500$ m) were also discarded since here the angle between the beams approaches $180\,^\circ$.

### 2.1.2 PPI measurements

The lidar at position 2 measured 45 different azimutal positions over three different elevations, performing a $60\,^\circ$ sweep every 45 s, scanning in the westerly direction (240–$300\,^\circ$). The elevation angles were 0.27, 0.84 and $1.41\,^\circ$. The full trajectory lasted 145 s accounting for the 10 s that the scan needed to return to its initial position. For every azimuthal position, 156 range gates

from 100 to 8150 m (separated every 50 m) were measured. The horizontal wind speed was reconstructed for every single scan and range gate, resulting in a horizontal wind speed at each range gate and elevation every 145 s, so four measurements were performed within a 10-min period. Due to the low availability of data at long ranges when using a filtering threshold (Floors



**Table 1.** Measurement periods, positions and operational availability for all of the systems used in the analysis.

| Position | System | Easting (m)/ Northing (m) | Height AMSL (m) | Measurement | Start/End Day/Month | Data (h) | Operational availability (%) |
|---|---|---|---|---|---|---|---|
| 1 | Scanning lidar | 446,080.03 6,259,660.30 | 12.36 | dual-setup | 03/12-17/02 | 1250.4 | 70.62 |
| 2 | Scanning lidar | 445,915.64 6,261,837.49 | 26.38 | PPI | 26/11-17/02 | 1575.2 | 79.38 |
| 3 | Scanning lidar | 445,823.66 6,263,507.90 | 42.97 | dual-setup | 03/12-17/02 | 1289.4 | 71.04 |
| 6 | Buoy | 438,441.00 6,262,178.00 | 0.00 | SST | 04/11-11/01 | 1636.0 | 100.00 |
| 7 | | 440,616.00 6,262,085.00 | | | | | |
| 8 | Høvsøre mast | 447,642.00 6,255,431.00 | 0.32 | Sonic (100 m) | 01/11-29/02 | 2740.7 | 96.51 |
| | | | | Wind vane (100 m) | 01/11-29/02 | 2756.3 | 97.07 |
| | | | | Wind cup (100 m) | 01/11-29/02 | 2757.0 | 97.09 |
| | | | | Temperature sensor (100 m) | 01/11-29/02 | 2168.0 | 76.37 |

et al., 2016), a dynamic filter is applied to "rescue" LOSs as shown in Beck and Kühn (2017). For every 10-min period, the probability density function of the data is calculated using a 2D-histogram. Measurement points fulfilling a lower threshold of –26.5 dB and an upper threshold of –5 dB are considered. LOSs below the CNR lower threshold are still considered, if their local probability density lies within one standard deviation of the mean probability density. LOS measurements below –30 dB are

5    always discarded. A final visual checking is applied to remove outliers. In Fig. 3, a comparison of the two filtering techniques is presented. As shown in the range-CNR plot, for these data the use of a dynamic filter extends the range of measurements from 4.6 to 6 km. The availability of LOS measurements for the two filtering techniques is shown in Fig. 4. For a distance of 6000 m from the coast, the use of the dynamic filter increases the data availability from 33.65 to 73.29%.

A comparison of the wind speeds observed by the dual-setup and the PPI at their matching positions 5 km offshore can be

10    found in Floors et al. (2016). In general, the 10-min mean reconstructed wind speeds from the PPI shows a good agreement with the dual-setup ones, especially close to the coast. At further distances to the coast, higher mean differences are found. These are related to the different size of the measuring volume of the lidars, the inherent temporal and spatial variability of the wind speed and the distinct reconstruction methods. While the reconstruction in the PPI is performed with a sinusoidal fit of





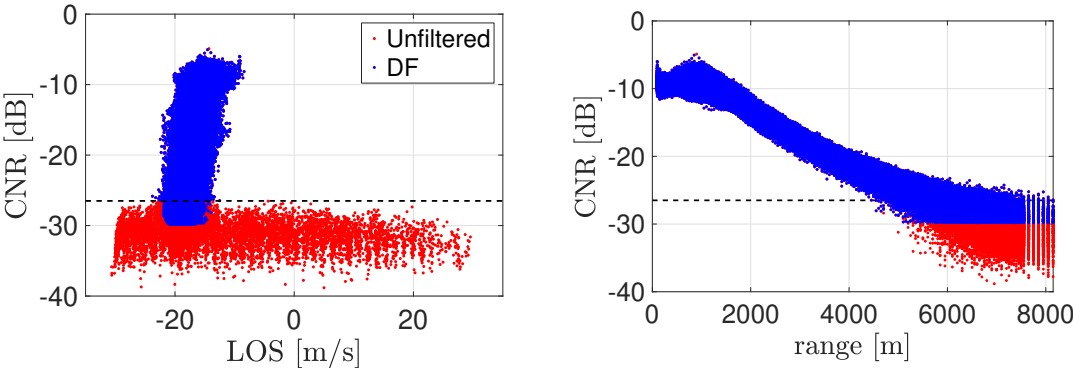

**Figure 3.** LOS-CNR plot (left) and range-CNR plot (right) for the original lidar data (both colors) and the data filtered with the dynamic filter (DF) (blue) for a 10-min measurement period. The dashed line represents the threshold line of -26.5 dB used in the conventional filter.

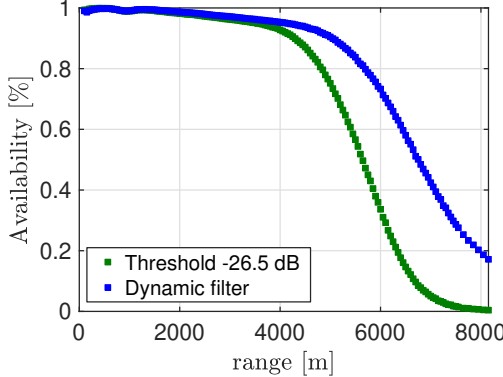

**Figure 4.** Availability of LOS measurements as a function of the range with a CNR threshold of -26.5 dB (green) and a dynamic filter (blue).

60°, the dual-setup uses two LOSs from the two lidars at a similar position in space. The uncertainties arising from the nature of the two systems are not clearly addressed. For the PPI, we need to assume horizontal flow homogeneity. At distant ranges wider areas are covered, and there is a higher uncertainty in the reconstruction. Besides, the PPI trajectories are not horizontal. For the dual-setup, we do not assume horizontal flow homogeneity, but the measurement ranges are longer than in the PPI.

## 5  3   Wind conditions

The campaign was characterized by strong south-westerly winds. For subsequent analysis we want to estimate the atmospheric stability conditions during the campaign. Since there are no measurements of this type in the offshore area, we estimate the offshore stability based on sonic anemometer measurements from the Høvsøre met-mast. We select the highest sonic anemometer, at 100 m, since this is the less influenced by the land effects and by internal boundary layers growing during




westerly winds. The analysis based on the derived Obukhov length $L$ at the 100 m sonic, reveals that during the winter months, there were mostly stable conditions (56.5%) followed by neutral (27.8%) and unstable (15.7%). Three classes are used for the stability classification, with $z/L < -0.1$ for unstable, $-0.1 \leq z/L \leq 0.1$ for neutral and $z/L > 0.1$ for stable conditions. To test if we can estimate the offshore stability based on the onshore measurements, we conduct a comparison of the gradient of the

potential temperature between the sea and the air, and the $L$ estimated from the sonic measurements. The directional wave buoy located at positions 6 and 7 measured the sea surface temperature every 30 min. Due to a major failure in the buoy system, only measurements until the beginning of January 2016 were recorded (see Table. 1 for more details). A comparison with the sea surface temperature (SST) derived from satellite images is shown in Fig. 5. The SST was computed from night-time observations from NOAA, AVHRR, Metop AVHRR, Terra MODIS, Aqua MODIS, Aqua AMSR-E, Envisat AATSR and MSG

Seviri satellites based on the interpolation method described in Høyer and She (2007). The spatial resolution of the satellite SST is $0.02° \times 0.02°$ and its temporal resolution is 24 h. Figure 5 shows that the differences between both SSTs are small and both SSTs do not vary as much as the air temperature measured by the temperature sensor at 100 m on the mast.

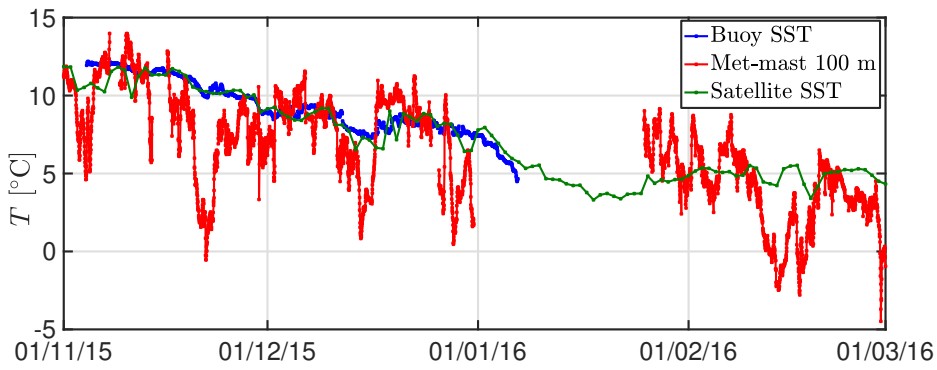

**Figure 5.** Time series of temparatures measured by the satellites, mast and buoy.

To conduct the comparison with the stability from the met-mast, we calculate the potential temperature gradient between the sea surface (buoy) and the air (met mast 100 m) for every 30-min period, first clustering the data according to the wind direction

and second using the 30-min averaged $L$ (sonic at 100 m). Only westerly and easterly sectors were analysed. The sign of the mean gradient of the potential temperature between the sea surface and the air for westerly winds (see Fig. 6) is in agreement with the stability from the sonic anemometer at 100 m. For easterly winds there is no such correspondence, as expected. For westerly winds we will assume that the stability measured by the onshore met-mast at 100 m is a good indicator of the stability of the offshore area. Our further analysis refers to data from westerly winds during neutral and stable conditions.

## 3.1   Coastal gradient for westerly winds

We analyse the influence of the land on the wind speed in the near-coastal area by using the dual-setup lidar observations at offshore positions. Due the reduced availability of measurements at distant positions, we look at 10-min periods up to 3 km





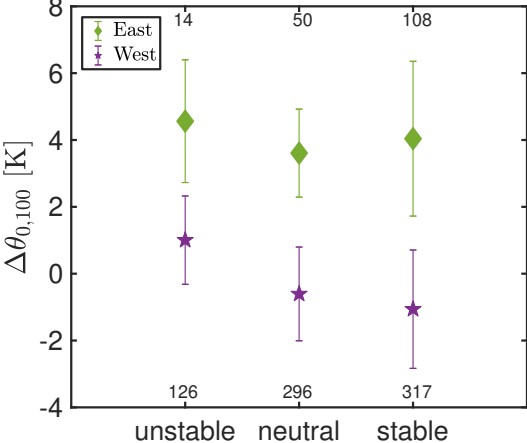

**Figure 6.** Potential temperature gradient between the SST and the air temperature at 100 m for three stability classes. The diamond (easterlies) and the star (westerlies) represent the mean and the error bars represent the standard deviation within each stability class. Numbers above and below the error bars refers to the number of periods used for the analysis.

offshore. Figure 7 shows the reconstructed 10-min mean wind speeds obtained from the dual-setup at 50, 100 and 150 m AMSL for periods with neutral stratification. For all heights, the flow slows down when approaching the coast.

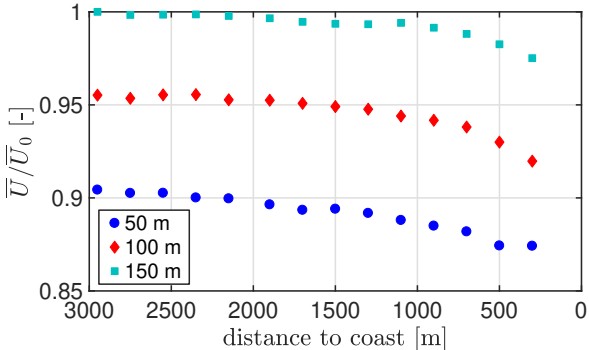

**Figure 7.** Normalized coastal wind gradient along the dual-setup transect of reconstructed wind speed measurements for westerly winds at 50, 100 and 150 m AMSL for neutral periods. $U_0$ is the mean wind speed at 2950 m to the coast at 150 m height.

## 4 Modelling coastal effects

We will use the PPI measurements further upstream of the coast to forecast winds at positions close to the coast where we also have PPI measurements. Our forecasting technique is first composed by an advection component, in which it is assumed that large turbulent structures are advected with the mean wind. Second we need to vertically extrapolate the wind because the





upstream PPI observations are at different heights than those closer to the coast. Last we need corrections due to the influence of the coast; as seen in Fig. 7 the wind has been observed as decreasing as it approaches the coast. Here, we will first show the way used to account for the coastal effects. This is done based on the dual-setup measurements as they are independent of PPI scans and are always performed at the same heights. In this section we will only use dual-setup measurements up to 3 km

during neutral conditions.

For a homogeneous and stationary flow, the mean wind speed profile is given as:

$$U(z) = \frac{u_*}{\kappa} \left[ \ln\left(\frac{z}{z_0}\right) - \Psi\left(\frac{z}{L}\right) \right] \tag{1}$$

where $U$ is the mean wind speed, $z$ the height above the ground, $u_*$ the friction velocity, $\kappa$ the von Kármán constant ($\approx 0.4$) and $z_0$ the roughness length. To account for stability effects $\Psi$ is included, which depends on the Obukhov length $L$. To model

the effects of the orography and roughness on the wind, which depend on the distance to the coast, we assume that the observed ($obs$) wind speed is:

$$U_{obs}(x,z) = U(z_0(x),z)O(x,z) \tag{2}$$

where $O$ is an orography correction that depends on the height and the distance to the coast $x$. Note that we assume that $z_0$ varies with the distance to the coast.

## 4.1   Orography effects

The orography effects are estimated using the microscale IBZ model, which is part of the Wind Atlas Analysis and Application Program (WAsP) (Troen and Lundtang Petersen, 1989). The orography correction was determined at each position measured by the dual-setup and for all wind directions using as an input a digital terrain model (Geostyrelsen, 2016). In Fig. 8 the orography corrections for the positions 500 and 2950 m from the coast and for all wind directions are shown. For westerly winds, at 500 m

from the coast and 50 m AMSL, the wind speed slows down $\approx 2\,\%$. On the other hand, for northerly and southerly winds the wind speeds up due to the presence of the cliff. The effects are reduced the further from the coast and with increasing height.

## 4.2   Roughness effects

We model the influence of the wind on the roughness of the water using the expression of Charnock (1955),

$$z_0 = \alpha_c \frac{u_*^2}{g}, \tag{3}$$

where $\alpha_c$ is the Charnock parameter and $g$ the acceleration due to gravity. For open ocean $\alpha_c = 0.011$ has been reported (Smith, 1980) while for near-coastal area, values between 0.008 and 0.06 can be found (Kraus, 1972). To determine the roughness length dependency with distance to shore we apply the following strategy. Once the dual-setup observations at the different range gates are corrected by using the orography corrections, these are used together with Eqs. (1) and (3) to determine both $u_*$ and $z_0$, and thus $\alpha_c$. Figure 9 left shows the dependency of the estimated roughness length with distance to the coast after



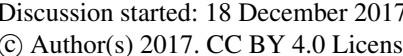

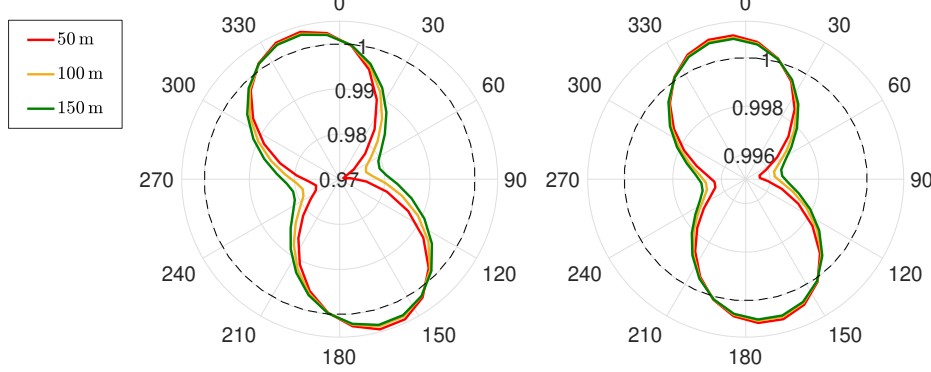

**Figure 8.** Directional orography effects at 500 m (top) and 2950 m (bottom) from the coast at the heights of 50, 100 and 150 m AMSL. Note the different scales in the plots.

applying the orography corrections for the neutral cases. The roughness length decreases with distance from the coast. Without orography corrections, the roughness length is slightly higher than the case with corrections close to the coast, as expected. Since the roughness length varies with distance to the coast, so does the Charnock parameter (see Fig. 9 right).

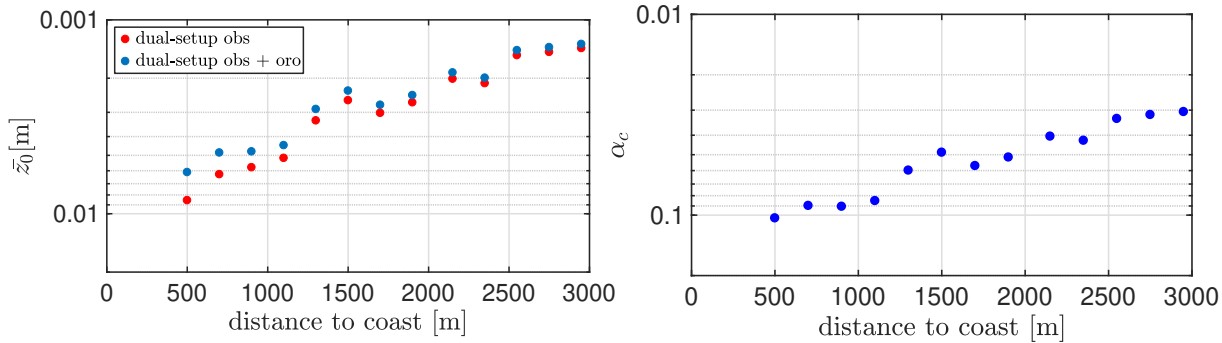

**Figure 9.** Mean estimated roughness length dependency with distance to the coast for the dual-setup mean wind profiles after and before applying orography corrections (left). Charnock's parameter dependency with the distance to the coast (right).

We test the estimated Charnock parameter dependency with distance to shore by selecting 10-min periods during neutral conditions where both PPI and dual-setup measurements were performed simultaneously. We fit Eqs. (1) and (3) to the orography corrected PPI measurements at those positions where we estimated the $\alpha_c$ dependency on distance to the coast with the dual-setup measurements. The comparison of the estimation of the wind using Eqs. (1) and (3) is shown in Fig. 10. As shown, with increasing distance to the coast, there is an increasing deviation of the fit from Eqs. (1) and (3) to the data, especially at the lowest height observations.





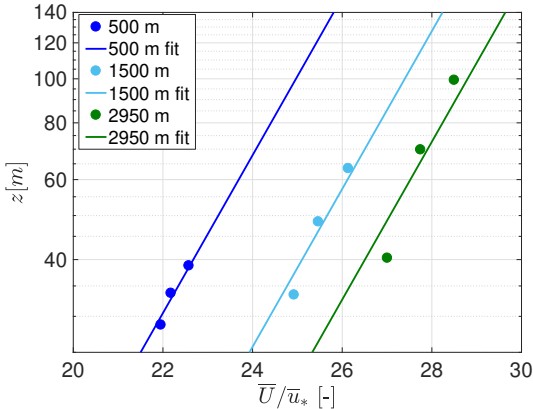

**Figure 10.** Comparison of the PPI observations with Eqs. (1) and (3) using the $\alpha_c$ dependency on distance to the coast at positions 500, 1500 and 2950 m from the coast.

## 5 Very short-term wind speed forecast

As mentioned earlier, we want to forecast wind speeds in a very short-term horizon by assuming Taylor's frozen turbulence hypothesis. For this purpose we consider two positions: the upstream position (1) and the downstream position or forecasting position (2), with the wind blowing from (1) to (2). If at a time $t$ a considerable change in wind speed occurs in the velocity of the position (1), this event will appear at the position (2) after some time $\Delta t$. In other words, this event can be foreseen at position (2) with a time ahead $\Delta t$. In our analysis, the downstream position is set to 500 m from the lidar at $z_2 = 33.76$ m, which corresponds to the height of the intermediate PPI elevation scan. Lidar measurements are performed at multiple upstream positions (range gates) from which the forecast can be originated. To keep a fixed forecast horizon, the upstream position (1) and height $z_1$, from which the wind is advected, are determined dynamically at every position in time using the moving-average wind speed at the downstream position. This can be understood as having multiple virtual met-masts over several distances from the downstream position. Because high wind speeds were observed during the measurement campaign, and the limit for high quality PPI measurements is $\approx 6$ km, we establish a forecast horizon of 5 min. We assume that a change in wind speed, observed 5 min ahead at the position (1) will propagate with a wind speed $v_1(t)$ and cover the distance $r_{12}(t)$ after the time delay $\Delta t$ of 5 min. But because $v_1(t)$ and $r_{12}$ might be not parallel, we use its projection on wind direction given by:

$$\boldsymbol{r_{12}}(t) \cdot \frac{\boldsymbol{v_1}(t)}{v_1(t)} = \Delta t v_1(t). \tag{4}$$

To incorporate the local effects and the changes in height between the upstream and downstream positions, we consider corrections in the wind speed due to height, roughness and orography. To evaluate the appropriateness of those corrections, we will compare a simple advection model (A), an advection model with height corrections (AH), an advection model with



**Table 2.** Computed statistics of wind speed (U), wind direction (dd), turbulence intensity (TI) and Obukhov Length (L) for all evaluated periods, based on the Høvsøre met-mast at 100 m.

| Period | Starting time | Duration | $\overline{U}$ [m/s] | $dd$ [°] | $\overline{TI}$ [%] | $\overline{L}$ [m] | stability class |
|---|---|---|---|---|---|---|---|
| 1 | 02.12.2015 04:20 | 6 h 10 min | 13.50 | [239-260] | 4.21 | 268 | stable |
| 2 | 04.12.2015 12:50 | 7 h 50 min | 16.56 | [229-275] | 6.60 | 2217 | neutral |
| 3 | 06.12.2015 19:10 | 6 h 20 min | 17.18 | [277-288] | 7.39 | -1640 | neutral |
| 4 | 09.12.2015 10:10 | 3 h 40 min | 12.89 | [240-273] | 6.36 | -4560 | neutral |
| 5 | 10.12.2015 22:50 | 5 h 10 min | 13.37 | [242-261] | 6.97 | 1383 | neutral |
| 6 | 12.12.2015 10:50 | 4 h 50 min | 8.15 | [256-304] | 5.91 | 166 | stable |
| 7 | 23.12.2015 15:50 | 7 h 30 min | 15.89 | [238-269] | 6.98 | 2178 | neutral |
| 8 | 27.12.2015 10:50 | 6 h 50 min | 16.34 | [248-282] | 7.38 | -10122 | neutral |
| 9 | 25.01.2016 23:50 | 8 h | 14.28 | [225-290] | 3.17 | 128 | stable |
| 10 | 31.01.2016 23:00 | 5 h 20 min | 7.48 | [250-301] | 3.80 | 37 | stable |

height and roughness length correction (AHR) and, finally, an advection model with corrections due to height, roughness length change and orography changes (AHRO). We evaluate our forecasting method against the well-known persistence method and an ARIMA model. A summary of the time periods in which the very short-term forecasting method is applied is shown in Table. 2. We select periods with wind speeds below 17 m/s and westerly periods, with a minimum duration of 3 h and with high availability of the data. No unstable periods fulfilled this criteria, therefore we focus here on neutral and stable conditions.

### 5.1 Advection model (A)

For the advection model, $U_2(t)$ is estimated as follows:

1. The upstream position (1) at $(x_1, z_1)$ from the PPI scan is determined dynamically using the 5-min moving average wind speed at the downstream position and the forecast time horizon $k$ (here 5 min). The wind direction from the previous forecasted step is used to calculate the projected distance from which the forecast is originated. For the positions in time and space in which observations at the upstream position are missing, the previous observation is used.

2. The observed wind speed at the upstream position is therefore advected, which means that the forecasted wind speed at the downstream position is considered to be the same as the wind speed in the upstream position, $U_{2,z2}(t) = U_{1,z1}(t-k)$

### 5.2 Advection model with height correction (AH)

This is similar as the A model but the wind speed is extrapolated to match the height of the downstream observation. To do so, following steps are followed:





1. Step 1 from model A is conducted.

2. The logarithmic profile in Eq. (1) is fit to three consecutive PPI wind speed observations at the position (1). The friction velocity $u_{*,1}$ and a roughness length $z_{0,1}$ are thus estimated.

3. The roughness length $z_{0,1}$ is used to correct the advected wind speed to the downstream height by using:

$$U_{2,z2}(t) = U_{1,z1}(t-k) \frac{\ln(z_1/z_{0,1}(t-k))}{\ln(z_2/z_{0,1}(t-k))}. \tag{5}$$

### 5.3 Advection model with height and roughness correction (AHR)

1. Steps 1 and 2 from the AH model are conducted.

2. The friction velocity $u_{*,1}$ and the roughness length $z_{0,1}$ are used to calculate the geostrophic wind at position (1)

$$G_1(t) = \frac{u_{*,1}(t)}{\kappa} \sqrt{\left(\ln\left(\frac{u_{*,1}(t)}{f z_{0,1}}\right) - A\right)^2 + B^2}, \tag{6}$$

where $f$ refers to the Coriolis parameter and $A = 1.8$, $B = 4.5$. We assume here that the geostrophic wind at the position (2) is the same as at the position (1):

$$G_2(t) = G_1(t-k). \tag{7}$$

3. The geosthropic wind is used to estimate the roughness length $z_{0,2}$ and the friction velocity $u_{*,2}$. To solve for both parameters in position (1), we assume a fixed Charnock parameter derived from the dual-setup analysis (see Fig. 9 right):

$$G_2(t) = \frac{u_{*,2}(t-k)}{\kappa} \sqrt{\left(\ln\left(\frac{g}{f \alpha_{c,2} u_{*,2}(t-k)}\right) - A\right)^2 + B^2}, \tag{8}$$

$$z_{0,2} = \alpha_{c,2} \frac{u_{*,2}^2}{g}. \tag{9}$$

4. The forecasted wind speed is:

$$U_{2,z2}(t) = \frac{u_{*,2}(t-k)}{\kappa} \ln\left(\frac{z_2}{z_{0,2}(t-k)}\right). \tag{10}$$

### 5.4 Advection model with height, roughness and orography correction (AHRO)

1. It is assumed that the corrections due to orography at positions further away are negligible.

2. Steps 1–3 from the AHR model are conducted.

3. The orography corrections at the downstream position are applied, i.e.

$$U_{2,z2}^o(t) = U_{2,z2}(t) \cdot O(x_2, z_2). \tag{11}$$





## 5.5 Statistical models

To evaluate the goodness of the forecasting techniques in Sect. 5, we use the benchmarks persistence and ARIMA models.

- ARIMA: denoted as ARIMA$(p, d, q)$ is a statistical model widely used in very short-term predictions of wind speeds (Kavasseri and Seetharaman, 2009). It uses recent past values (Autoregressive (AR)) and recent residuals of the forecast (Moving Average (MA)) to predict current values. This model is suitable to analyse non-stationary processes, since it uses non-seasonal differences ($d$) to build the forecasting model. A general equation for the ARIMA model is:

$$U(t+k) = \sum_{i=1}^{p} \Phi_i U(t+k-i) + a(t+k) - \sum_{j=1}^{q} \Theta_j a(t+k-j), \qquad (12)$$

where $\Phi_i$ is the i-th autoregressive parameter, $\Theta_j$ is the j-th moving average parameter, $a(t)$ is the error term at time $t$, $k$ is the forecasting horizon and $U(t)$ is the value of the wind speed observed at the time $t$. Here we build a new ARIMA model for each period. To test the stationarity of the time-series, we first look at its autocorrelation function (ACF) and its partial autocorrelation (PACF). If it has positive autocorrelations out to a high number of lags, we include an order of differentiation $d$. To test if this order is sufficient, we look at the residuals of the differentiated time-series and perform a unit root test using the Dickey-Fuller test (Dickey and Fuller, 1979). To determine the order $p$ and $q$ of the ARIMA model, we compute the ACF and the PACF of the stationarized time series, following the method explained in Cadenas and Rivera (2007). The model chosen for every period is the one which minimises the residuals. For every individual set, the previous hour of observations is used to derive the AR and MA parameters using the method by Box and Jenkins (1976).

- Persistence: This is a particular case of the ARIMA model in which $q = 0$, $p = 1$, $d = 0$ and the AR coefficient is set to 1, since it assumes that the previous and the current value are high correlated. Our predicted wind speed is defined as:

$$U(t) = U(t-k). \qquad (13)$$

## 6 Results

We evaluate the accuracy of the 5-min forecast of wind speeds based on the described advection techniques, and compare it with the results of the statistical methods persistence and ARIMA. To do so, three criteria are employed namely, the Root Mean Square Error (RMSE), the Mean Bias Error (MBE) and the Maximum Absolute Error (MaxAE). Table 3 includes the RMSE, MBE and MaxAE for all periods stated in Table 2. Minimum values are indicated in bold.

For neutral conditions (periods 2, 3, 4, 5, 7 and 8), the advection model with corrections performs in general better than the statistical forecasting models. As an example, the distribution of errors produced by all models for period 7 can be seen in Fig. 11. The forecasting error $\epsilon$ is defined as $\epsilon_i = U_{p,i} - U_{ob,i}$, where $U_{ob,i}$ is the actual observation for a time position $t_i$ and $U_{p,i}$ is the forecast for the same period. The statistical methods show a broader distribution of errors. This is because ARIMA and persistence fail to predict the phase of the events, since they construct their predictions according to the previous observations.





**Table 3.** RMSE, MBE and MaxAE statistics for all periods evaluated.

| Period | Stability | | A | AH | AHR | AHRO | P | ARIMA | *p,d,q* parameters |
|---|---|---|---|---|---|---|---|---|---|
| | | RMSE (m/s) | 2.69 | 1.84 | 1.26 | 0.99 | 0.49 | **0.44** | |
| 1 | stable | MBE (m/s) | 2.64 | 1.78 | 1.19 | 0.90 | **-0.01** | -0.04 | 3,1,0 |
| | | MaxAE (m/s) | 4.12 | 2.91 | 2.28 | 1.95 | **1.21** | 1.54 | |
| | | RMSE (m/s) | 2.61 | 1.29 | 0.75 | **0.71** | 1.01 | 0.93 | |
| 2 | neutral | MBE (m/s) | 2.48 | 1.08 | 0.26 | -0.10 | **-0.01** | -0.15 | 3,1,1 |
| | | MaxAE (m/s) | 4.64 | 2.71 | **2.29** | 2.65 | 3.29 | 2.87 | |
| | | RMSE (m/s) | 2.16 | **0.87** | 0.91 | 1.05 | 1.10 | 0.91 | |
| 3 | neutral | MBE (m/s) | 1.95 | 0.37 | -0.48 | -0.73 | **0.04** | 0.34 | 2,0,1 |
| | | MaxAE (m/s) | 4.10 | **2.25** | 2.87 | 3.04 | 3.22 | 2.63 | |
| | | RMSE (m/s) | 1.51 | 0.68 | **0.59** | 0.74 | 0.81 | 0.73 | |
| 4 | neutral | MBE (m/s) | 1.36 | 0.37 | -0.19 | -0.49 | **-0.09** | -0.29 | 1,0,1 |
| | | MaxAE (m/s) | 2.79 | **1.49** | 1.55 | 1.83 | 2.03 | 1.75 | |
| | | RMSE (m/s) | 1.81 | 0.93 | **0.70** | 0.76 | 1.13 | 0.97 | |
| 5 | neutral | MBE (m/s) | 1.56 | 0.60 | **0.01** | -0.29 | 0.05 | 0.34 | 1,0,0 |
| | | MaxAE (m/s) | 4.42 | 3.09 | 2.41 | **2.09** | 3.21 | 2.22 | |
| | | RMSE (m/s) | 0.78 | 0.53 | **0.52** | 0.54 | 0.70 | 0.72 | |
| 6 | stable | MBE (m/s) | 0.53 | 0.18 | -0.08 | -0.20 | 0.05 | **0.01** | 1,1,1 |
| | | MaxAE (m/s) | 2.11 | **2.01** | 2.30 | 2.31 | 2.67 | 2.78 | |
| | | RMSE (m/s) | 2.33 | 1.15 | **0.90** | 0.97 | 1.20 | 1.16 | |
| 7 | neutral | MBE (m/s) | 2.10 | 0.74 | **-0.02** | -0.37 | 0.07 | 0.33 | 2,0,0 |
| | | MaxAE (m/s) | 5.39 | **2.95** | 3.89 | 4.23 | 3.69 | 2.92 | |
| | | RMSE (m/s) | 2.62 | 1.15 | **0.79** | 0.87 | 1.02 | 0.96 | |
| 8 | neutral | MBE (m/s) | 2.45 | 0.83 | -0.03 | -0.37 | **-0.02** | -0.01 | 2,1,1 |
| | | MaxAE (m/s) | 4.54 | 2.73 | **1.98** | 2.20 | 2.95 | 2.65 | |
| | | RMSE (m/s) | 3.01 | 2.22 | 1.62 | 1.36 | 0.43 | **0.44** | |
| 9 | stable | MBE (m/s) | 2.96 | 2.16 | 1.55 | 1.28 | **0.01** | 0.07 | 1,0,1 |
| | | MaxAE (m/s) | 4.52 | 3.43 | 2.62 | 2.27 | 1.39 | **1.30** | |
| | | RMSE (m/s) | 0.44 | 0.34 | **0.42** | 0.48 | 0.43 | 0.45 | |
| 10 | stable | MBE (m/s) | 0.20 | -0.06 | -0.26 | -0.35 | **0.04** | 0.22 | 1,0,0 |
| | | MaxAE (m/s) | 1.39 | **1.00** | 1.21 | 1.28 | 1.46 | 1.10 | |

For the neutral periods 4, 5, 7 and 8, the forecasting accuracy of the AHR model is higher than that of any other advection model. In those periods, introducing the orography correction results in an underestimation of the wind speed, as it can be seen in the MBE of those periods. The dependency of the forecasting errors on the mean wind speed of the downstream observation

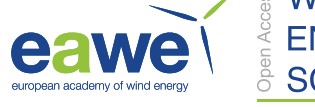



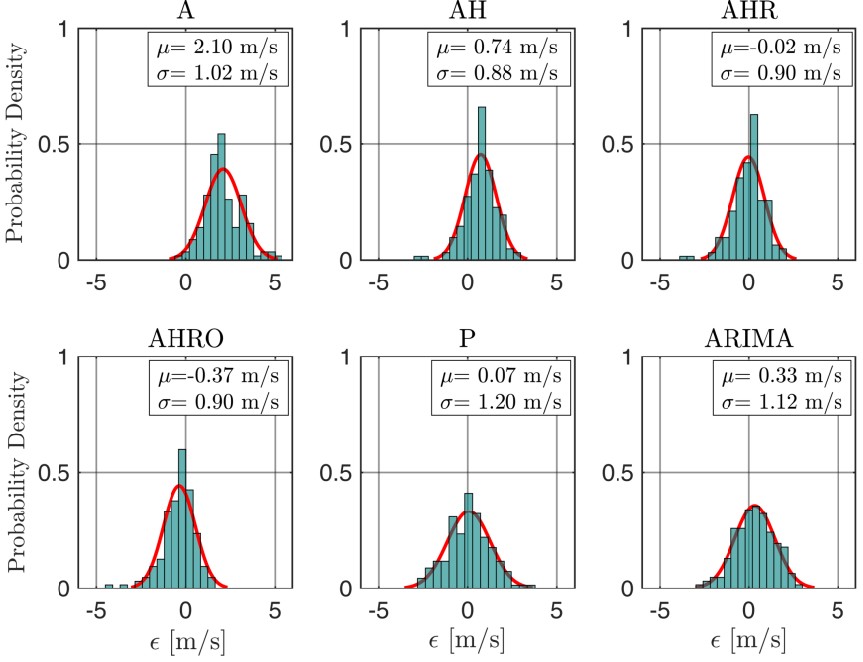

**Figure 11.** Histogram of the forecast errors $\epsilon$ for period 7 (neutral) for all evaluated models. The red line represents a normal distribution with the same mean $\mu$ and standard deviation $\sigma$ as the distribution of errors.

is shown in Fig. 12. For wind speeds close to 16 m/s and neutral conditions, AHRO produces smaller errors (Fig. 12 left). Therefore for period 2, which has a higher mean wind speed, introducing the orography correction results in a more accurate forecast than that of any other model. This is because the roughness change correction is estimated with mean neutral profiles, whose mean wind speed at the forecasting height is also close to 16 m/s. For period 3, the one with the highest wind speed, the

increasing underprediction of AHRO and AHR with wind speed results in AH predicting better than the other models.

When looking at the forecast of wind speeds during period 7 (see Fig. 13), we can see that the advection models are able to forecast the phase of the events, but the forecast does not contain as many fluctuations as the observed wind speed at the downstream position. To analyse if this is due to the model or to the nature of the observations, the dependency of the level of fluctuations on the horizontal reconstructed wind speeds with the distance of the measurements is investigated. In Fig. 14,

the ensemble average of the standard deviation of $U$, computed for every hour and elevation angle during periods where all measurements are available, is displayed. The standard deviation is higher close to the coast. This is because of the higher turbulence level close to the coast compared to positions further offshore, due to local effects, the difference in height in the observations and the reconstruction of horizontal wind speeds from the lidar. Since the availability of lidar observations decreases with the distance, the reconstruction of wind speeds acts as a low-pass filter.

For stable cases (periods 1, 6, 9 and 10) the performance of the advection-based models is quite similar to the performance seen in neutral cases. During periods 6 and 10, the AHR model produces smaller errors than the statistical models. Figure 15

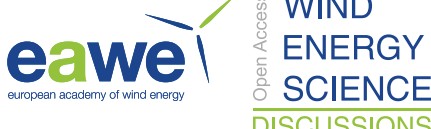



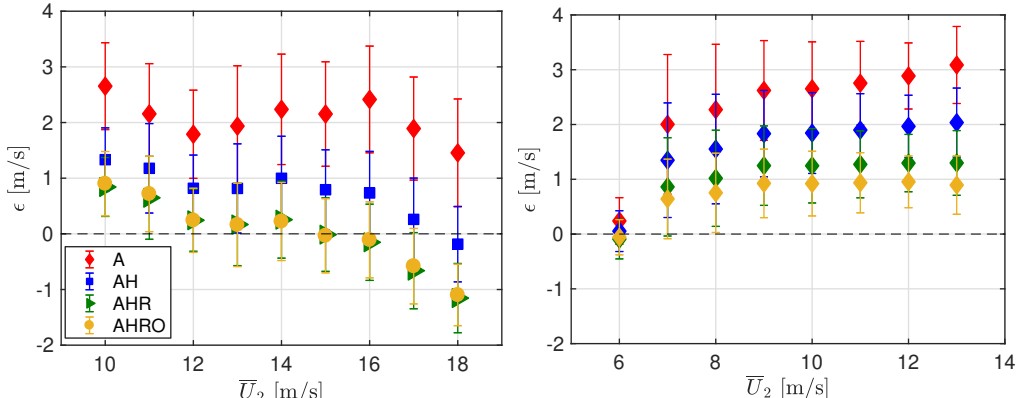

**Figure 12.** Forecasting error dependency on wind speed for the advection models A, AH, AHR and AHRO for neutral (left) and stable (right) periods.

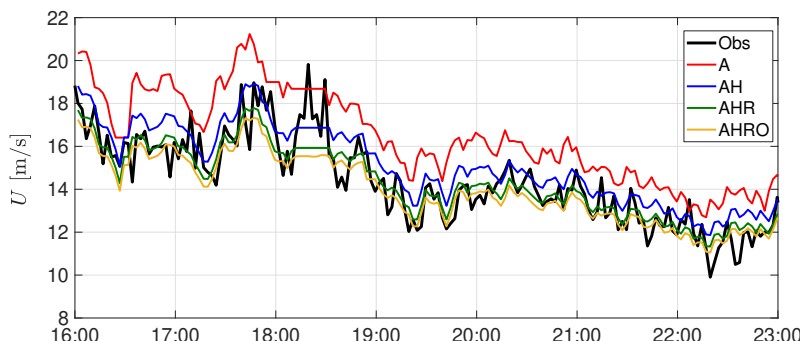

**Figure 13.** Time series of wind speed observations (Obs) and predictions with the A, AH, AHR and AHRO models for period 7.

shows the comparison of the observed and forecasted wind speeds for all models during period 6. The figure shows that there is more scatter for the persistence and ARIMA models than for the advection models. For high wind speeds, this effect is also found in the advection models.

For periods 1 and 9, all advection models show larger errors than the conventional models. This is because these periods are characterized by higher wind speeds than periods 6 and 10. The effect of the mean wind speed in the forecasting error of stable cases is shown in Fig. 12 right. Above 6 m/s the forecast error tends to increase with wind speed. For higher wind speeds, the forecast originates from further upstream positions and consequently higher heights. If we now look at the differences between the PPI observations at 2950 m and the estimation of wind speeds using Eqs. (1) and (3) from the dual-setup observations (see Fig. 10), we can see that the differences are more pronounced at further distances. Thus, it is difficult to accurately predict the magnitude of the wind speed during stable conditions and high wind speeds, due to the increasing height in the observations at further positions, the differences in the dual-setup and PPI observations and the assumption of neutral stabilities during stable

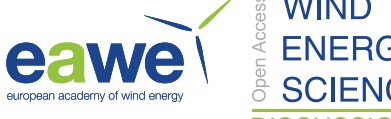


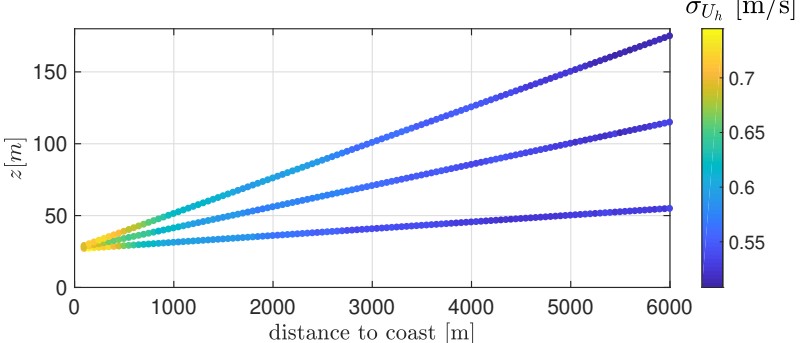

**Figure 14.** Ensemble average standard deviation of the horizontal wind speed with distance to shore for the three elevations angles during all periods analysed.

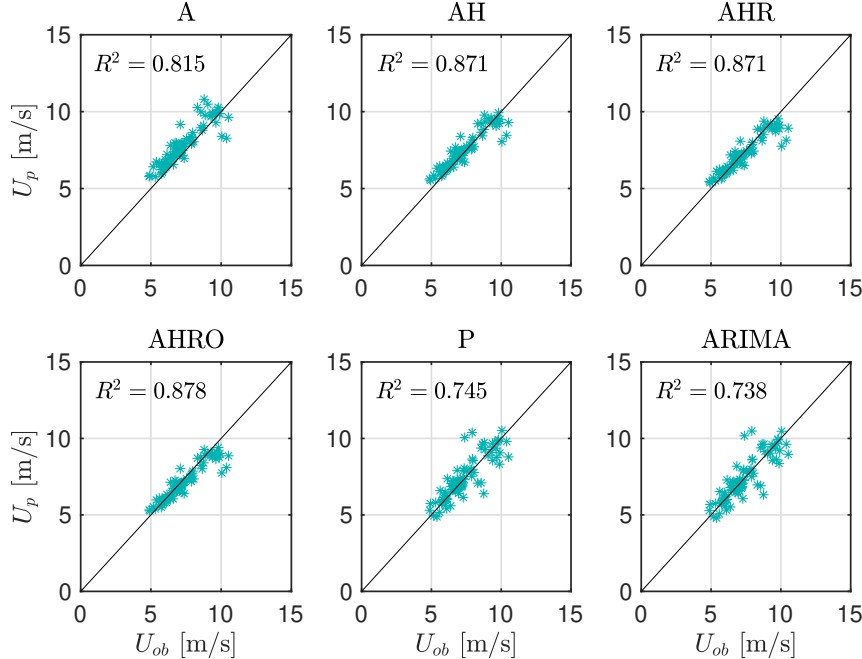

**Figure 15.** Comparison between the observed $ob$ and predicted $p$ wind speed for all evaluated models for period 6 (stable) with N=95.

conditions. However, in quantifying the errors for both stable and neutral cases, the RMSE of the stable cases is in general smaller than those of the neutral periods, because during stable conditions the inflow is less turbulent.



## 7 Concluding remarks

This paper evaluated the use of wind lidar observations for very short-term forecast of near-coastal winds. From our analysis on periods with neutral atmospheric conditions, in all cases, the advection-based models with corrections were able to predict the wind speeds better than the benchmarks persistence and ARIMA. Due to the different turbulent conditions experienced at

every range position, the forecasting technique was not able to predict the turbulence of the fluctuations. This is partly due to the presence of the coast, the wind speed reconstruction using lidar measurements and the measurement itself, acting as a low-pass filter at further distances. During stable periods, we could only produce an accurate prediction of the magnitude of the wind speed during low wind speeds. This is because of the increasing height in the observations at further positions, the differences in the dual-setup and PPI observations and the assumption of neutral stability during stable conditions, due to a

lack of a precise estimate of the stability offshore.

The corrections applied in our advection-based models to forecast the magnitude of the wind speed observation are necessary due to the tilted trajectories and local effects of the coastline and the cliff. However, the corrections are not perfect. The results are based on a limited amount of dual-setup measurements and it is clear that we could not find a model with a zero mean bias error. The best performing advection model depends on the wind speed and stability. Despite all these limitations, we

showed that lidars i) provide range-resolved information to derive site-specific effects influencing the wind speed and ii) are promising candidates for very short-term wind power applications since they can forecast wind speeds with more accuracy than the benchmarks persistence and ARIMA. To use an advection-based wind speed forecasting technique, one could better benefit from horizontal trajectories that do not require height corrections. Additionally, applying this technique in pure offshore areas improves the results, since no corrections due to local effects are required. An operational lidar forecasting system on an

offshore wind farm would need no corrections at all.

As very short-term wind power forecasts typically use statistical techniques that learn from the wind speed and power data at the location of interest and surroundings, based on our results, a long-range lidar system is likely to decrease the uncertainty in the prediction of offshore wind power, especially during ramp events, where the statistical methods do not perform well.

Further research will focus on using long-range, remotely sensed wind speed observations to predict the power produced by

a single wind turbine or a wind farm.

*Author contributions.* Laura Valldecabres conducted the research work and wrote the paper. Alfredo Peña and Michael Courtney extensively contributed to the modelling of the coastal effects and the use of lidar observations for forecasting wind speed respectively. Lueder von Bremen and Martin Kühn supervised the research work and contributed to the structure of the paper. All co-authors participated in the outline and review of the manuscript.

*Competing interests.* The authors declare that they have no conflict of interest.

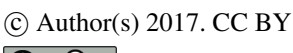



*Acknowledgements.* This project has received funding from the European Union's Horizon 2020 research and innovation programme under the Marie Sklodowska-Curie grant agreement No 642108. Funding from the ForskEL program to the project "RUNE "No. 12263 and from the Ministry of Science and Culture of Lower Saxony to the project "ventus efficiens "(ZN3024, MWK Hannover) are acknowledged.





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
