# Peer review of "Very short-term forecast of near-coastal flow using scanning lidars"

_Wind Energy Science, 2017_

## Referee Comment (RC1) · Anonymous Referee #1 · 7 Jan 2018

By analyzing data from the so-called RUNE experiment, the authors demonstrate in this piece of work that wind speeds at a downstream position can be forecast by using measurements from a scanning lidar performed upstream in a very short-term horizon. This topic relates to a very interesting and innovative (and at the same time relevant) field of work, and this is why I would rate also the scientific significance of the paper as rather high. Also the scientific quality is good – all steps of the investigations are explained in sufficient detail and quality. What I am missing – this as a more general comment – is a further discussion of the forecast horizon in relation to the range of the lidar measurements. The authors explain why they only could use data with a reach of up to 6 km. But I would think that is rather a weak point of the specific experiment, and data with higher ranges (possibly up to 10 km or beyond) may be available for

very near-future analyses. How would this impact the results of the study? A further general comment relates to the structure of the manuscript, which I think should be overworked by the authors. The sections 'Wind data analysis', 'Wind conditions' and 'Modelling coastal effects' amount to a major part of the paper before the actual key part (the section on the forecasting itself) is reached – I do not think that the proportions are fair here, and I am also missing the central theme at some points. Furthermore, I think the partitioning of the individual sections may be revised – the description at the beginning of section 2 is e.g. followed by sub-section 2.1 (and two sub-subsections) but no further sub-section. Sections 4 and 5 then have another structure. Please check again carefully if the structure of the manuscript really supports the logical chain and development of the argumentation or if this can be improved.

Some minor comments – in the order of their appearance in the document:

[p.1 l.10] A figure from 2015 is given here – but the manuscript is from end of 2017. Please try to find a more up-to-date figure.

[p.3 first paragraph] Readers who do not know about the RUNE experiment already may miss that RUNE is the name of a publicly funded project run by DTU and partners. Please add these details. Also the Hovsore test site may not be known by all readers.

[p.3 Figure 1] I would prefer to have the explanations of the numbers/positions (only type of measurement system maybe) in the caption.

[p.3 l.10] Here it says that position 6 and 7 are for a short-range lidar – which is actually a floating lidar – but in the remainder of the text it is only referred to the data from a wave buoy. This needs to be clarified. Was the wave buoy deployed at the same position (twice) as the floating lidar?

[p.4 Figure 2] Figure needs to be reworked. For instance, I can see only one black line – and also details are not easy to be depicted.

[p.4 l.10] Here it says that 'Observations close to the lidar systems were also discarded

since here the angle between the beams approaches 180°'. This should be explained in some more detail (why this is bad), and the beams' geometry may be shown to define a certain threshold.

[p.8 l.1] ... 'shows the [averaged] reconstructed 10-min-mean wind speeds', I guess – this should be explained/specified in some more detail.

[p.10 Figure 8] 'top' and 'bottom' in the caption should be left and right I guess – please correct, and/or add identifications (a) and (b) or similar. Beyond, it is rather difficult to read and understand the figures – please add some more explanation and also make the scales better comparable.

[p.11 eq.(4)] This needs to be explained/specified further – I guess the bold letters refer to vector quantities (?)

[p.12 l.4] Here it says that only 'periods with wind speeds below 17 m/s' were selected, but this is not the case for period 3. Please comment on this or correct statement, respectively.

[p.19 l.4] ... 'were able to predict the wind speeds better than the benchmarks'. Please quantify this better (for your conclusions). Can you estimate the corresponding impact on a possible application?

[p.22 ll.20-21] The reference seems not to be complete, please add details. Is this an article?

Thanks again for this very interesting and informative manuscript – I am looking forward to seeing a revised version.

---

## Referee Comment (RC2) · Anonymous Referee #2 · 21 Jan 2018

I found the organization of the paper followed a logical progression and clearly described the derivation of the corrections, models forecasts, and model evaluations. The well-organized structure made the paper easy to follow and was overall an enjoyable paper to read.

The demonstrated use of lidar measurements and the evaluation of applications of short-term forecasting models along a coast with variable height is a topical and valuable topic of study to the scientific community and applied wind energy development community.

I agree in general with the comments of the prior reviewer.

Technical comments:

[Figure]

[p.3 l.7] The position number, 8, of the mast could be specified here for clarity.

[p.3 Figure 1] The right-hand scale should be labeled, presumably "height [m]".

[p.17 l.2] It is not clear what is referred to by "this effect." Presumably it means equivalent scatter in the advection results relative to the persistence and ARIMA models. Please clarify.

[p.18 l.1] This conclusion that the RMSE is smaller for stable cases than for neutral cases was not immediately apparent from the full set of results in the table. If this is referring to a comparison of the errors from the best fit models for each case, rather than the full set of results, please clarify that.

Minor typographical comments:

[p.2 l.16] I believe the comma belongs before the word "especially" rather than after it.

[p.2 l.18] No comma is needed after the word "Oregon"

[p.3 l.16] The word "see" should be "sea"

[p.4 l.12] The word "azimutal" should be spelled "azimuthal" in English.

[p.7 Figure 5] The word "temparatures" should be spelled "temperatures" in English.

[p.7 l.22] The word "to" should be added between the words "Due the"

[p.9 l.2] The word "method" is recommended to replace the word "way"

[p.11 l.14] To keep the subject of the sentence consistent for clarity, it is recommended to replace the first phrase with "But because r12 might not be parallel with v1(t)"

[p.14 l.23] I believe the comma belongs before the word "namely" rather than after it.

[p.15 l.2] The word "it" should be removed from "as it can"

---

## Author Response (AR1)

Dear Editor:

Thank you very much for the opportunity to address the comments from the reviewers to our manuscript. The reviewers' comments significantly improved the quality of our document. The responses to their comments have been submitted separated to each reviewer. We also modified/add some sentences in the Results and Concluding remarks sections to improve the understanding of the paper. The implemented changes are listed below in blue and highlighted in the marked-up manuscript. Page and lines refer to the original document.

Sincerely,

The authors

**Responses to Referee #1**

What I am missing – this as a more general comment – is a further discussion of the forecast horizon in relation to the range of the lidar measurements. The authors explain why they only could use data with a reach of up to 6 km. But I would think that is rather a weak point of the specific experiment, and data with higher ranges (possibly up to 10 km or beyond) may be available for very near-future analyses. How would this impact the results of the study?

Thanks for the interesting point. We have now included a paragraph regarding this comment in the Concluding Remarks section, between the first and second original paragraphs.

In this paper the forecasting horizon is limited to 5 min due to the maximum range of the lidar measurements (6 km) and the high wind speeds experienced during the measurement campaign. A long-range lidar system with a maximum range of 10 km could forecast wind speeds up to 17 m/s, thus generating forecasts with a horizon of 10 min. Since commercially available ultra-range lidars can now measure up to 30 km (Kameyama et al., 2012) the forecasting horizon for this application could be extended up to 30 min ahead.

Kameyama, S., Sakimura, T., Watanabe, Y., Ando, T., Asaka, K., Tanaka, H., Yanagisawa, T., Hirano, Y., and Inokuchi, H.: Wind sensing demonstration of more than 30km measurable range with a 1.5mu;m coherent Doppler lidar which has the laser amplifier using Er,Yb:glass planar waveguide, in: Proc.SPIE, vol. 8526, pp. 8526 − 8526 − 6, https://doi.org/10.1117/12.977330, 2012.

A further general comment relates to the structure of the manuscript, which I think should be overworked by the authors. The sections 'Wind data analysis', 'Wind conditions' and 'Modelling coastal effects' amount to a major part of the paper before the actual key part (the section on the forecasting itself) is reached – I do not think that the proportions are fair here, and I am also missing the central theme at some points. Furthermore, I think the partitioning of the individual sections may be revised – the description at the beginning of section 2 is e.g. followed by sub-section 2.1 (and two sub-subsections) but no further sub-section. Sections 4 and 5 then have another structure. Please check again carefully if the structure of the manuscript really supports the logical chain and development of the argumentation or if this can be improved.

This part has been improved by including the wind conditions section (previously 3) as a subsection into the wind data section (now 2.2). The section Modelling coastal effects (previously section 4 and now section 3) has a new title "Modelling coastal effects for wind speed forecasting correction" to clarify its purpose. We believe it is necessary to explain in detail how we modelled the coastal effects. Therefore, we consider this part a full section of the paper.

[p.1 l.10] A figure from 2015 is given here – but the manuscript is from end of 2017. Please try to find a more up-to-date figure.

This has been updated with the new data released from the year 2017. A new record set by Denmark of 43.4%. Thanks for the recommendation.

In 2017 Denmark produced a record 43.4% of the country's electricity with wind energy.

Reference:

The Danish Wind Industry Association (DWIA): Wind energy production as a percentage of total electricity consumtion 2005 - 2017, http://www.windpower.org/en/knowledge/statistics/the_danish_market.html, 2018.

**[p.3 first paragraph] Readers who do not know about the RUNE experiment already may miss that RUNE is the name of a publicly funded project run by DTU and partners. Please add these details. Also the Hovsore test site may not be known by all readers.**

The authors consider they have included enough references to the project. Besides they think that the information about the project should be included in the Acknowledgement part, as it is done.

Regarding the Hovsore test site, this part has been modified:

Our study is based on measurements performed during the Reducing Uncertainty of Near-shore wind resource Estimates (RUNE) campaign (Simon and Courtney, 2016; Floors et al., 2016). The experiment was conducted at the western coast of Denmark, north of the area of Høvsøre (see Fig. 1) and close to one of DTU's wind turbine test station. A comprehensive analysis of the wind conditions at Høvsøre during a ten years period from the test station's meteorological mast, located 1.7 km east of the North Sea (see Fig. 1, position 8) is presented in Peña et al. (2016).

**[p.3 Figure 1] I would prefer to have the explanations of the numbers/positions (only type of measurement system maybe) in the caption.**

We have changed the caption of the figure as suggested:

Figure 1: Map of the area of the RUNE campaign indicating the positions of the dual-setup lidars (1 and 3), the PPI lidar (2), the profiling lidars (2, 4, 5, 6 (7)) the met-mast (8) and the wave buoy (9) located 150 m away from position 6.

**[p.3 l.10] Here it says that position 6 and 7 are for a short-range lidar – which is actually a floating lidar – but in the remainder of the text it is only referred to the data from a wave buoy. This needs to be clarified. Was the wave buoy deployed at the same position (twice) as the floating lidar?**

The wave buoy was located at a different position (position 9). This has been updated in the table and corrected throughout the document. Two references for the wave buoy have also been added.

Sanchez, R. and Rørbæk, K.: Metocean Buoy Deployment, Tech. rep., DHI, 2016.

Floors, R., Lea, G., Pena Diaz, A., Karagali, I., and Ahsbahs, T.: Report on RUNE's coastal experiment and first inter-comparisons between measurements systems, Tech. rep., DTU Wind Energy E-0115(EN), DTU Wind Energy: Roskilde, Denmark, 2016a.

**[p.4 Figure 2] Figure needs to be reworked. For instance, I can see only one black line – and also details are not easy to be depicted.**

Figure 2 has been reworked for clarity purposes.

[Figure]

**[p.4 l.10]** Here it says that 'Observations close to the lidar systems were also discarded some more detail (why this is bad), and the beams' geometry may be shown to define a certain threshold.

The angle between the beams should be large enough to allow measuring a difference in radial speed, but if it gets close to 180° this could lead to an error in the reconstruction of the wind speed when the wind is perpendicular to the beam direction.

We have added the following sentence after "since here the angle between the beams approaches 180°" and reference to the text:

"and, consequently, the uncertainty of the reconstructed speed becomes very high (Stawiarski et al., 2013)."

Stawiarski, C., Träumner, K., Knigge, C., and Calhoun, R.: Scopes and Challenges of Dual-Doppler Lidar Wind Measurements—An Error Analysis, Journal of Atmospheric and Oceanic Technology, 30, 2044-2062, https://doi.org/10.1175/JTECH-D-12-00244.1, 2013

We also included in page 6, line 4, after "the measurement ranges are longer than in the PPI"

"and, consequently, the uncertainty in the sensing height will be higher."

**[p.8 l.1]** ... 'shows the [averaged] reconstructed 10-min-mean wind speeds', I guess – this should be explained/specified in some more detail.

This has been clarified in the new version:

Figure 7 shows the ensemble average wind speed of all 10-min mean wind speeds reconstructed from the dual-setup observations…

**[p.10 Figure 8] 'top' and 'bottom' in the caption should be left and right I guess – please correct, and/or add identifications (a) and (b) or similar. Beyond, it is rather difficult to read and understand the figures – please add some more explanation and also make the scales better comparable.**

This has been modified in the new version. For better comparison only the orography corrections for 50 and 150 m AMSL are shown. The same scale is used for both plots. The caption now says:

[Figure]

Figure 8: Directional orography effects O(x,z,dd) at x=500 m (left) and x=2950 m (right) from the coast at two heights AMSL.

We have also modified Eq. (2) on page 9, since the orography correction also depends on the wind direction.

$$U_{obs}(x, z, dd) = U(z_0(x), z)O(x, z, dd)$$

, where O is an orography correction that depends on the height, the distance to the coast x and the wind direction dd. Note that we assume that z0 varies with the distance to the coast.
Also in page 13 we have modified the following text:

The orography corrections at the downstream position are applied using the measured wind direction at (1), i.e.

$$U^o_{2,z2} = U_{2,z2}(t)O(x_2, z_2, dd_1)$$

**[p.11 eq.(4)] This needs to be explained/specified further – I guess the bold letters refer to vector quantities (?)**

The bold quantities refer to vector quantities. This section has been reformulated in the paper.

If at a time t a considerable change in wind speed occurs at position (1), this event will appear at the position (2) after some time Δt. In other words, this event can be foreseen at position (2) with a time ahead Δt. In our analysis, the downstream position is set to 500 m from the PPI lidar (position 2) in the westerly direction at z2= 33.76 m, which corresponds to the height of the intermediate PPI elevation scan. Lidar measurements are performed at multiple upstream positions (range gates) from which the forecast can be originated. This can be understood as having multiple virtual met-masts over several distances west of the downstream position. To keep a fixed forecast horizon, the upstream position (1) and height z1, from which the wind is advected, are determined dynamically at each time stamp using the 5-min moving average wind speed v2 (t) and direction at the downstream position. But because the vector v2(t) might not be parallel to the line

of virtual met-masts, we use the vector projection of the advected distance on the wind direction $|r12|=|\Delta tv2(t)|\cdot\cos(\theta)$, with $\theta$ defined as the angle between the wind direction and 270 °. Because high wind speeds were observed during the measurement campaign, and the limit for high quality PPI measurements is≈ 6 km, we establish a forecast horizon of 5 min. We assume that a change in wind speed, observed 5 min ahead at the position (1) will propagate with a wind speed $v1(t)$ and travel the distance $r12$ in the time $\Delta t= 5$ min.

**[p.12 l.4] Here it says that only 'periods with wind speeds below 17 m/s' were selected,**
**but this is not the case for period 3. Please comment on this or correct statement, respectively.**

We have corrected this in the new version.

…with mean wind speeds below 18 m/s.

**[p.19 l.4] … 'were able to predict the wind speeds better than the benchmarks'. Please**
**quantify this better (for your conclusions). Can you estimate the corresponding impact on a possible application?**

We have calculated the improvement over persistence and ARIMA for all advection models and included a table with the results. Table 4.

We also included the following paragraphs in the results section:

Page 14. L25. After "in bold".

The improvements of the advection models over the benchmarks persistence and ARIMA are shown in Table 4. Values corresponding to best performance are indicated in bold.

Page 14. L26. After "statistical forecasting models":
The improvement over persistence using the best calibrated advection model for each period, ranges from 21-38%. Compared to the benchmark ARIMA the improvement ranges from 4-28%.

Page 18. L2. After "less turbulent".
For stable cases, disregarding the periods of high wind speed (1 and 9), the best calibrated advection models give improvements over persistence of 21-26 % and over ARIMA of 24-28 %

**Table 4.** Improvement of all advection models over the benchmarks persistence ($\text{Imp}_P$) and ARIMA ($\text{Imp}_A$).

| Period | stability | A | | AH | | AHR | | AHRO | |
|---|---|---|---|---|---|---|---|---|---|
| | | $\text{Imp}_P$ (%) | $\text{Imp}_A$ (%) | $\text{Imp}_P$ (%) | $\text{Imp}_A$ (%) | $\text{Imp}_P$ (%) | $\text{Imp}_A$ (%) | $\text{Imp}_P$ (%) | $\text{Imp}_A$ (%) |
| 1 | stable | -448.98 | -511.36 | -275.51 | -318.18 | -157.14 | -186.36 | -102.04 | -125.00 |
| 2 | neutral | -158.42 | -180.65 | -27.72 | -38.71 | 25.74 | 19.35 | **29.70** | **23.66** |
| 3 | neutral | -96.36 | -137.36 | **20.91** | **4.40** | 17.27 | 0.01 | 4.55 | -15.38 |
| 4 | neutral | -86.42 | -106.85 | 16.05 | 6.85 | **27.16** | **19.18** | 8.64 | -1.37 |
| 5 | neutral | -60.18 | -86.60 | 17.70 | 4.12 | **38.05** | **27.84** | 32.74 | 21.65 |
| 6 | stable | -11.43 | -8.33 | 24.29 | 26.39 | **25.71** | **27.78** | 22.86 | 25.01 |
| 7 | neutral | -94.17 | -100.86 | 4.17 | 0.86 | **25.02** | **22.41** | 19.17 | 16.38 |
| 8 | neutral | -156.86 | -172.92 | -12.75 | -19.79 | **22.55** | **17.71** | 14.71 | 9.37 |
| 9 | stable | -600.02 | -584.09 | -416.28 | -404.55 | -276.74 | -268.18 | 216.28 | -209.09 |
| 10 | stable | -2.33 | 2.22 | **20.93** | **24.44** | 2.33 | 6.67 | -11.63 | -6.67 |

In the Concluding remarks section we have also included the following paragraphs.

At the beginning:
This paper evaluated the use of wind lidar observations for very short-term forecast of near-coastal winds. From our analysis on periods with neutral atmospheric conditions, the best fitted advection-based model with corrections showed an improvement over the benchmarks persistence and ARIMA of 21-38% and 4-28%, respectively.

After "original line 23":
Our analysis is a first input component to a decision-making model that may include spot market prices, scheduled supply and demand and balancing costs. Thus, here it is not intended to quantify the economic impact of using a lidar-based wind speed forecast. However, as the balancing costs are proportional to the root mean square error, it can be assumed that they will decrease. In particular, as in most of the periods analysed the maximum absolute error is lower than that of the benchmarks, using a lidar-based wind speed forecast might have a positive impact on integrating offshore wind power into the grid.

**[p.22 ll.20-21] The reference seems not to be complete, please add details. Is this an article?**
This reference (technical report) is now completed.

**Responses to Referee #2**

[p.3 l.7] The position number, 8, of the mast could be specified here for clarity.

We have now specified it.

[p.3 Figure 1] The right-hand scale should be labeled, presumably "height [m]".

We have added this to the caption.

[p.17 l.2] It is not clear what is referred to by "this effect." Presumably it means equivalent scatter in the advection results relative to the persistence and ARIMA models.
Please clarify.

We have modified the text in the new version:

For wind speeds above 8 m/s a high scatter between the advection models and the observations is also found.

[p.18 l.1] This conclusion that the RMSE is smaller for stable cases than for neutral cases was not immediately apparent from the full set of results in the table. If this is referring to a comparison of the errors from the best fit models for each case, rather than the full set of results, please clarify that.

This has been modified in the new version as follows:

However, in quantifying the errors for the best fitted advection model in both stable and neutral cases, the RMSE of the stable cases is in general smaller than those of the neutral periods, because during stable conditions the inflow is less turbulent.

Minor typographical comments:

[p.2 l.16] I believe the comma belongs before the word "especially" rather than after it.
This has been implemented in the new version.

[p.2 l.18] No comma is needed after the word "Oregon"
This has been implemented in the new version.

[p.3 l.16] The word "see" should be "sea"
This has been implemented in the new version.

[p.4 l.12] The word "azimutal" should be spelled "azimuthal" in English.
This has been implemented in the new version.

[p.7 Figure 5] The word "temparatures" should be spelled "temperatures" in English.
This has been implemented in the new version.

[p.7 l.22] The word "to" should be added between the words "Due the"
This has been implemented in the new version.

[p.9 l.2] The word "method" is recommended to replace the word "way"
This has been implemented in the new version.

[p.11 l.14] To keep the subject of the sentence consistent for clarity, it is recommended to replace the first phrase with "But because r12 might not be parallel with v1(t)"
This paragraph has been reformulated. Please see Reply to Referee 1.

[p.14 l.23] I believe the comma belongs before the word "namely" rather than after it.
This has been implemented in the new version.

[p.15 l.2] The word "it" should be removed from "as it can"
This has been implemented in the new version.

**Other changes**

Page 8. Line 6
We replaced: "composed by" with "based on".

Page14. Line 30
After "to the previous observations" we included:
The dependency of the forecasting errors on the mean wind speed of the downstream observation for all advection models is shown in Fig. 12. From there it can be inferred that the orography correction is required, since the AHR model overestimates wind speeds in the range of 10 to 17 m/s.

We later removed the sentence in line 3 page 15: "The dependency of the forecasting errors on the mean wind speed…"

Page16 Line 3
We replaced "This is because the roughness change correction is estimated" with "This is because both the roughness change correction and the orography corrections are estimated"

Page 16 Line 6.
The paragraph "When looking at the forecast" has been removed from this section and placed after the analysis of the results for stable periods.

*Page 18 Line 1.*
We included the following sentence after "conditions"
Although we include the shear in our advection models we are not considering the atmospheric stability.

Page 18 Line 2.
After this paragraph we included the paragraph from Page 16 Line 6, which has also been modified for clarification.

When looking at the forecast of wind speeds during period 7 (see Fig. 14), we can see that the advection models are able to forecast the phase of the events, but the forecast does not contain as many fluctuations as the observed wind speed at the downstream position. To analyse if this is due to the model or to the nature of the observations, the dependency of the level of fluctuations on the horizontal reconstructed wind speeds with the distance of the measurements is investigated. In Fig. 15, the ensemble average of the standard deviation of U, computed for every hour and elevation angle during periods where all measurements are available, is displayed. The standard deviation observed by the lidar is higher the closer to the coast. We attribute this to a combination of two sources: site-specific conditions and measurement artifacts. In the first source we consider the higher roughness length close to the coast, compared to positions further offshore, and the topographic effects. In the second source we include the different height in the observations for the different ranges and the different arc length used for the reconstruction of horizontal wind speeds from the lidar. Since the arc length used for the measurement increases with the distance, the reconstruction of wind speeds acts as a low-pass filter for further distances. This filtering effect deteriorates the prediction of the magnitude of the events, and consequently influences the maximum absolute error.

Page 19 Line 1
We added:
This paper evaluated the use of wind lidar observations for very short-term forecast of near-coastal winds, using wind speed advection-based models.

Page 19 Line 5.
We replaced:
"This is partly due to the presence of the coast, the wind speed reconstruction using lidar measurements and the measurement itself, acting as a low-pass filter at further distances" with "We attribute these differences partly due to the presence of the coast increasing the turbulence level as the flow approaches and the low-pass filtering inherent in the wind speed reconstruction from the lidar measurements".

Page 19 Line 8
We replaced:

"This is because of the increasing height in the observations at further positions, the differences in the dual-setup and PPI observations and the assumption of neutral stability during stable conditions, due to a lack of a precise estimate of the stability offshore " with "This is a reflection of the increasing difficulty of predicting winds as i) the observations height increase at further positions ii) the differences in the dual-setup and PPI observations and iii) the assumption of neutral stability during stable conditions, due to a lack of a precise estimate of the offshore stability".

Page 19 Line 19
We replaced observation with "observations"

Page 19 Line 20
We included:
Thus, it is reasonable to expect that the forecasting performance of such a system would be better than the best results we have achieved since the many corrections might not have benefited the forecasting accuracy.

Page 19 Line 21
We replaced:
"As very short-term wind power forecasts typically use statistical techniques that learn from the wind speed and power data at the location of interest and surroundings, based on our results, a long-range lidar system is likely to decrease the uncertainty in the prediction of offshore wind power, especially during ramp events, where the statistical methods do not perform well" with " Very short-term wind power forecasts typically use statistical techniques that learn from the wind speed and power data at the location of interest and surroundings. Based on our results, a long-range lidar system is likely to decrease the uncertainty in the prediction of offshore wind power, especially during ramp events, i.e. large variation in wind speed within a short period of time, where statistical methods do not perform well.

[revised manuscript text omitted]
  dual-setup lidars (1 and 3), the PPI lidar (2), the profiling lidars (2, 4, 5, 6 (7)) the met-mast (8) and the wave buoy (9) located 150 m away from position 6. The colorbar shows the height above mean sea level in meters.

During the RUNE campaign, which took place during November 2015 to February 2016, profiling and scanning lidars were deployed to measure near-coastal wind conditions (see Fig.1, position 8). Four short-range (positions 2, 4, 5 and 6 (later 7)), and one long-range (position 2) profiling lidars measured the wind profile. One scanning lidar (position 2) was operated in Plan Position Indicator (PPI) mode, also known as the "sector-scan" scenario. Simultaneously, two more scanning lidars

15   (at positions 1 and 3) were configured in a dual trajectory to match at positions along three horizontal virtual lines. In what follows we will refer to them as the dual-setup. In Fig. 2 the positions of the dual-setup and the PPI are shown. The PPI and the dual-setup trajectories were designed so that the measurements will intersect at 5000 m offshore at 50, 100 and 150 m AMSL. Further, a directional wave buoy (position 9) was deployed to measure waves, currents and  sea

[revised manuscript text omitted]
 offshore. Figure 7 shows the  ensemble average wind speed of all 10-min mean wind speeds  reconstructed from the dual-setup observations at 50, 100 and 150 m AMSL for periods with neutral stratification. For all heights, the flow slows down when approaching the coast.

[Figure]

**Figure 7.** Normalized coastal wind gradient along the dual-setup transect of reconstructed wind speed measurements for westerly winds at 50, 100 and 150 m AMSL for neutral periods. $U_0$ is the mean wind speed at 2950 m to the coast at 150 m height.

**3 Modelling coastal effects for wind speed forecasting correction**

We will use the PPI measurements further upstream of the coast to forecast winds at positions close to the coast where we also have PPI measurements. Our forecasting technique is first  based on an advection component, in which it is assumed that large turbulent structures are advected with the mean wind. Second we need to vertically extrapolate the wind

5 because the upstream PPI observations are at different heights than those closer to the coast. Last we need corrections due to the influence of the coast; as seen in Fig. 7 the wind has been observed as decreasing as it approaches the coast. Here, we will first show the  method used to account for the coastal effects. This is done based on the dual-setup measurements as they are independent of PPI scans and are always performed at the same heights. In this section we will only use dual-setup measurements up to 3 km during neutral conditions.

10 For a homogeneous and stationary flow, the mean wind speed profile is given as:

$$U(z) = \frac{u_*}{\kappa} \left[ \ln\left(\frac{z}{z_0}\right) - \Psi\left(\frac{z}{L}\right) \right] \tag{1}$$

where $U$ is the mean wind speed, $z$ the height above the ground, $u_*$ the friction velocity, $\kappa$ the von Kármán constant ($\approx 0.4$) and $z_0$ the roughness length. To account for stability effects $\Psi$ is included, which depends on the Obukhov length $L$. To model the effects of the orography and roughness on the wind, which depend on the distance to the coast, we assume that the observed

15 ($obs$) wind speed is:

$$U_{obs}(x, z, dd) = U(z_0(x), z)O(x, z, dd) \tag{2}$$

where $O$ is an orography correction that depends on the height  $z$, the distance to the coast $x$ and the wind direction $dd$. Note that we assume that $z_0$ varies with the distance to the coast.

**3.1 Orography effects**

The orography effects are estimated using the microscale IBZ model, which is part of the Wind Atlas Analysis and Application Program (WAsP) (Troen and Lundtang Petersen, 1989). The orography correction was determined at each position measured by the dual-setup and for all wind directions using as an input a digital terrain model (Geostyrelsen, 2016). In Fig. 8 the orography corrections for the positions 500 and 2950 m from the coast and for all wind directions are shown. For westerly winds, at 500 m from the coast and 50 m AMSL, the wind speed slows down $\approx 2\,\%$. On the other hand, for northerly and southerly winds the wind speeds up due to the presence of the cliff. The effects are reduced the further from the coast and with increasing height.

[Figure]

**Figure 8.** Directional orography effects $O(x,z,dd)$ at $x$=500 m (left) and $x$=2950 m (right) from the coast at  two heights  AMSL.

**3.2 Roughness effects**

We model the influence of the wind on the roughness of the water using the expression of Charnock (1955),

$$z_0 = \alpha_c \frac{u_*^2}{g}, \tag{3}$$

where $\alpha_c$ is the Charnock parameter and $g$ the acceleration due to gravity. For open ocean $\alpha_c = 0.011$ has been reported (Smith, 1980) while for near-coastal area, values between 0.008 and 0.06 can be found (Kraus, 1972). To determine the roughness length dependency with distance to shore we apply the following strategy. Once the dual-setup observations at the different range gates are corrected by using the orography corrections, these are used together with Eqs. (1) and (3) to determine both $u_*$ and $z_0$, and thus $\alpha_c$. Figure 9 left shows the dependency of the estimated roughness length with distance to the coast after applying the orography corrections for the neutral cases. The roughness length decreases with distance from the coast. Without orography corrections, the roughness length is slightly higher than the case with corrections close to the coast, as expected. Since the roughness length varies with distance to the coast, so does the Charnock parameter (see Fig. 9 right).

[Figure]

**Figure 9.** Mean estimated roughness length dependency with distance to the coast for the dual-setup mean wind profiles after and before applying orography corrections (left). Charnock's parameter dependency with the distance to the coast (right).

We test the estimated Charnock parameter dependency with distance to shore by selecting 10-min periods during neutral conditions where both PPI and dual-setup measurements were performed simultaneously. We fit Eqs. (1) and (3) to the orography corrected PPI measurements at those positions where we estimated the $\alpha_c$ dependency on distance to the coast with the dual-setup measurements. The comparison of the estimation of the wind using Eqs. (1) and (3) is shown in Fig. 10. As shown, with increasing distance to the coast, there is an increasing deviation of the fit from Eqs. (1) and (3) to the data, especially at the lowest height observations.

[Figure]

**Figure 10.** Comparison of the PPI observations with Eqs. (1) and (3) using the $\alpha_c$ dependency on distance to the coast at positions 500, 1500 and 2950 m from the coast.

**4 Very short-term wind speed forecast**

As mentioned earlier, we want to forecast wind speeds in a very short-term horizon by assuming Taylor's frozen turbulence hypothesis. For this purpose we consider two positions: the upstream position (1) and the downstream position or forecasting position (2), with the wind blowing from (1) to (2). If at a time $t$ a considerable change in wind speed occurs  at position (1), this event will appear at the position (2) after some time $\Delta t$. In other words, this event can be foreseen at position (2) with a time ahead $\Delta t$. In our analysis, the downstream position is set to 500 m from the  PPI lidar (position 2) in the westerly direction at $z_2 = 33.76$ m, which corresponds to the height of the intermediate PPI elevation scan. Lidar measurements are performed at multiple upstream positions (range gates) from which the forecast can be originated. This can be understood as having multiple virtual met-masts over several distances west from the downstream position. To keep a fixed forecast horizon, the upstream position (1) and height $z_1$, from which the wind is advected, are determined dynamically at  each time stamp using the 5-min moving-average wind speed $\overline{v_2}(t)$ and direction at the downstream position.  But because the vector $\overline{v_2}(t)$ might not be parallel to the line of virtual met-masts , we use the vector projection of the advected distance on the wind direction $|\boldsymbol{r_{12}}| = |\Delta t \overline{v_2}(t)| \cdot \cos(\theta)$, with $\theta$ defined as the angle between the wind direction and $270\,°$. 
[revised manuscript text omitted]

For neutral conditions (periods 2, 3, 4, 5, 7 and 8), the advection model with corrections performs in general better than the statistical forecasting models. The improvement over persistence using the best calibrated advection model for each period, ranges from 21-38% (see Table 4). Compared to the benchmark ARIMA the improvement ranges from 4-28%. As an example, the distribution of errors produced by all models for period 7 can be seen in Fig. 11. The forecasting error $\epsilon$ is defined as $\epsilon_i = U_{p,i} - U_{ob,i}$, where $U_{ob,i}$ is the actual observation for a time position $t_i$ and $U_{p,i}$ is the forecast for the same period. The

**Table 3.** RMSE, MBE and MaxAE statistics for all periods evaluated.

| Period | Stability | | A | AH | AHR | AHRO | P | ARIMA | *p,d,q* parameters |
|---|---|---|---|---|---|---|---|---|---|
| 1 | stable | RMSE (m/s) | 2.69 | 1.84 | 1.26 | 0.99 | 0.49 | **0.44** | |
| | | MBE (m/s) | 2.64 | 1.78 | 1.19 | 0.90 | **-0.01** | -0.04 | 3,1,0 |
| | | MaxAE (m/s) | 4.12 | 2.91 | 2.28 | 1.95 | **1.21** | 1.54 | |
| 2 | neutral | RMSE (m/s) | 2.61 | 1.29 | 0.75 | **0.71** | 1.01 | 0.93 | |
| | | MBE (m/s) | 2.48 | 1.08 | 0.26 | -0.10 | **-0.01** | -0.15 | 3,1,1 |
| | | MaxAE (m/s) | 4.64 | 2.71 | **2.29** | 2.65 | 3.29 | 2.87 | |
| 3 | neutral | RMSE (m/s) | 2.16 | **0.87** | 0.91 | 1.05 | 1.10 | 0.91 | |
| | | MBE (m/s) | 1.95 | 0.37 | -0.48 | -0.73 | **0.04** | 0.34 | 2,0,1 |
| | | MaxAE (m/s) | 4.10 | **2.25** | 2.87 | 3.04 | 3.22 | 2.63 | |
| 4 | neutral | RMSE (m/s) | 1.51 | 0.68 | **0.59** | 0.74 | 0.81 | 0.73 | |
| | | MBE (m/s) | 1.36 | 0.37 | -0.19 | -0.49 | **-0.09** | -0.29 | 1,0,1 |
| | | MaxAE (m/s) | 2.79 | **1.49** | 1.55 | 1.83 | 2.03 | 1.75 | |
| 5 | neutral | RMSE (m/s) | 1.81 | 0.93 | **0.70** | 0.76 | 1.13 | 0.97 | |
| | | MBE (m/s) | 1.56 | 0.60 | **0.01** | -0.29 | 0.05 | 0.34 | 1,0,0 |
| | | MaxAE (m/s) | 4.42 | 3.09 | 2.41 | **2.09** | 3.21 | 2.22 | |
| 6 | stable | RMSE (m/s) | 0.78 | 0.53 | **0.52** | 0.54 | 0.70 | 0.72 | |
| | | MBE (m/s) | 0.53 | 0.18 | -0.08 | -0.20 | 0.05 | **0.01** | 1,1,1 |
| | | MaxAE (m/s) | 2.11 | **2.01** | 2.30 | 2.31 | 2.67 | 2.78 | |
| 7 | neutral | RMSE (m/s) | 2.33 | 1.15 | **0.90** | 0.97 | 1.20 | 1.16 | |
| | | MBE (m/s) | 2.10 | 0.74 | **-0.02** | -0.37 | 0.07 | 0.33 | 2,0,0 |
| | | MaxAE (m/s) | 5.39 |  2.95 | 3.89 | 4.23 | 3.69 | **2.92** | |
| 8 | neutral | RMSE (m/s) | 2.62 | 1.15 | **0.79** | 0.87 | 1.02 | 0.96 | |
| | | MBE (m/s) | 2.45 | 0.83 | -0.03 | -0.37 | **-0.02** | -0.01 | 2,1,1 |
| | | MaxAE (m/s) | 4.54 | 2.73 | **1.98** | 2.20 | 2.95 | 2.65 | |
| 9 | stable | RMSE (m/s) | 3.01 | 2.22 | 1.62 | 1.36 | 0.43 | **0.44** | |
| | | MBE (m/s) | 2.96 | 2.16 | 1.55 | 1.28 | **0.01** | 0.07 | 1,0,1 |
| | | MaxAE (m/s) | 4.52 | 3.43 | 2.62 | 2.27 | 1.39 | **1.30** | |
| 10 | stable | RMSE (m/s) | 0.44 | 0.34 | **0.42** | 0.48 | 0.43 | 0.45 | |
| | | MBE (m/s) | 0.20 | -0.06 | -0.26 | -0.35 | **0.04** | 0.22 | 1,0,0 |
| | | MaxAE (m/s) | 1.39 | **1.00** | 1.21 | 1.28 | 1.46 | 1.10 | |

statistical methods show a broader distribution of errors. This is because ARIMA and persistence fail to predict the phase of the events, since they construct their predictions according to the previous observations.

**Table 4.** Improvement of all advection models over the benchmarks persistence ($\text{Imp}_P$) and ARIMA ($\text{Imp}_A$).

| | | A | | AH | | AHR | | AHRO | |
|---|---|---|---|---|---|---|---|---|---|
| Period | stability | $\text{Imp}_P$ (%) | $\text{Imp}_A$ (%) | $\text{Imp}_P$ (%) | $\text{Imp}_A$ (%) | $\text{Imp}_P$ (%) | $\text{Imp}_A$ (%) | $\text{Imp}_P$ (%) | $\text{Imp}_A$ (%) |
| 1 | stable | -448.98 | -511.36 | -275.51 | -318.18 | -157.14 | -186.36 | -102.04 | -125.00 |
| 2 | neutral | -158.42 | -180.65 | -27.72 | -38.71 | 25.74 | 19.35 | **29.70** | **23.66** |
| 3 | neutral | -96.36 | -137.36 | **20.91** | **4.40** | 17.27 | 0.01 | 4.55 | -15.38 |
| 4 | neutral | -86.42 | -106.85 | 16.05 | 6.85 | **27.16** | **19.18** | 8.64 | -1.37 |
| 5 | neutral | -60.18 | -86.60 | 17.70 | 4.12 | **38.05** | **27.84** | 32.74 | 21.65 |
| 6 | stable | -11.43 | -8.33 | 24.29 | 26.39 | **25.71** | **27.78** | 22.86 | 25.01 |
| 7 | neutral | -94.17 | -100.86 | 4.17 | 0.86 | **25.02** | **22.41** | 19.17 | 16.38 |
| 8 | neutral | -156.86 | -172.92 | -12.75 | -19.79 | **22.55** | **17.71** | 14.71 | 9.37 |
| 9 | stable | -600.02 | -584.09 | -416.28 | -404.55 | -276.74 | -268.18 | 216.28 | -209.09 |
| 10 | stable | -2.33 | 2.22 | **20.93** | **24.44** | 2.33 | 6.67 | -11.63 | -6.67 |

[Figure]

**Figure 11.** Histogram of the forecast errors $\epsilon$ for period 7 (neutral) for all evaluated models. The red line represents a normal distribution with the same mean $\mu$ and standard deviation $\sigma$ as the distribution of errors.

The dependency of the forecasting errors on the mean wind speed of the downstream observation for all advection models is shown in Fig. 12. From there it can be inferred that the orography correction is required, since the AHR model overestimates wind speeds in the range of 10 to 17 m/s. For the neutral periods 4, 5, 7 and 8, the forecasting accuracy of the AHR model is higher than that of any other advection model. In those periods, introducing the orography correction results in an underestimation of the wind speed, as  can be seen in the MBE of those periods.  For wind speeds close to 16 m/s and neutral conditions, AHRO produces smaller errors (Fig. 12 left). Therefore for period 2, which has a higher mean wind speed, introducing the orography correction results in a more accurate forecast than that of any other model. This is because both the roughness change correction

5     and the orography corrections are estimated with mean neutral profiles, whose mean wind speed at the forecasting height is also close to 16 m/s. For period 3, the one with the highest wind speed, the increasing underprediction of AHRO and AHR with wind speed results in AH predicting better than the other models.

[Figure]

**Figure 12.** Forecasting error dependency on wind speed for the advection models A, AH, AHR and AHRO for neutral (left) and stable (right) periods.

10

15

20   For stable cases (periods 1, 6, 9 and 10) the performance of the advection-based models is quite similar to the performance seen in neutral cases. During periods 6 and 10, the AHR model produces smaller errors than the statistical models. Figure 13 shows the comparison of the observed and forecasted wind speeds for all models during period 6. The figure shows that there

is more scatter for the persistence and ARIMA models than for the advection models. For  wind speeds above 8 m/s a high scatter between the advection models and the observations is also found.

[Figure]

**Figure 13.** Comparison between the observed $ob$ and predicted $p$ wind speed for all evaluated models for period 6 (stable) with N=95.

For periods 1 and 9, all advection models show larger errors than the conventional models. This is because these periods are characterized by higher wind speeds than periods 6 and 10. The effect of the mean wind speed in the forecasting error of stable cases is shown in Fig. 12 right. Above 6 m/s the forecast error tends to increase with wind speed. For higher wind speeds, the forecast originates from further upstream positions and consequently higher heights. If we now look at the differences between the PPI observations at 2950 m and the estimation of wind speeds using Eqs. (1) and (3) from the dual-setup observations (see Fig. 10), we can see that the differences are more pronounced at further distances. Thus, it is difficult to accurately predict the magnitude of the wind speed during stable conditions and high wind speeds, due to the increasing height in the observations at further positions, the differences in the dual-setup and PPI observations and the assumption of neutral stabilities during stable conditions. Although we include the shear in our advection models we are not considering the atmospheric stability.

However, in quantifying the errors for the best fitted advection model in both stable and neutral cases, the RMSE of the stable cases is in general smaller than those of the neutral periods, because during stable conditions the inflow is less turbulent. For stable cases, disregarding the periods of high wind speed (1 and 9), the best calibrated advection models give improvements over persistence of 21-26 % and over ARIMA of 24-28 %.

When looking at the forecast of wind speeds during period 7 (see Fig. 14), we can see that the advection models are able to forecast the phase of the events, but the forecast does not contain as many fluctuations as the observed wind speed at the

downstream position. To analyse if this is due to the model or to the nature of the observations, the dependency of the level of fluctuations on the horizontal reconstructed wind speeds with the distance of the measurements is investigated. In Fig. 15, the ensemble average of the standard deviation of U, computed for every hour and elevation angle during periods where all measurements are available, is displayed. The standard deviation observed by the lidar is higher the closer to the coast.

5 We attribute this to a combination of two sources: site-specific conditions and measurement artifacts. In the first source we consider the higher roughness length close to the coast, compared to positions further offshore, and the topographic effects. In the second source we include the different height in the observations for the different ranges and the different arc length used for the reconstruction of horizontal wind speeds from the lidar. Since the arc length used for the measurement increases with the distance, the reconstruction of wind speeds acts as a low-pass filter for further distances. This filtering effect deteriorates

10 the prediction of the magnitude of the events, and consequently influences the maximum absolute error.

[Figure]

**Figure 14.** Time series of wind speed observations (Obs) and predictions with the A, AH, AHR and AHRO models for period 7.

[Figure]

**Figure 15.** Ensemble average standard deviation of the horizontal wind speed with distance to shore for the three elevations angles during all periods analysed.

**6 Concluding remarks**

This paper evaluated the use of wind lidar observations for very short-term forecast of near-coastal winds, using wind speed advection-based models. From our analysis on periods with neutral atmospheric conditions, the best fitted advection-based model with corrections showed an improvement over the benchmarks persistence and ARIMA of 21-38 % and 4-28 %, respectively. Due to the different turbulent conditions experienced at every range position, the forecasting technique was not able to predict the turbulence of the fluctuations. We attribute these differences partly due to the presence of the coast increasing the turbulence level as the flow approaches and the low-pass filtering inherent in the wind speed reconstruction from the lidar measurements. During stable periods, we could only produce an accurate prediction of the magnitude of the wind speed during low wind speeds. This is a reflection of the increasing difficulty of predicting winds as i) the observations height increase at further positions ii) the differences in the dual-setup and PPI observations and iii) the assumption of neutral stability during stable conditions, due to a lack of a precise estimate of the offshore stability.

In this paper the forecasting horizon is limited to 5 min due to the maximum range of the lidar measurements (6 km) and the high wind speeds experienced during the measurement campaign. A long-range lidar system with a maximum range of 10 km could forecast wind speeds up to 17 m/s, thus generating forecasts with a horizon of 10 min. Since commercially available ultra-range lidars can now measure up to 30 km (Kameyama et al., 2012) the forecasting horizon for this application could be extended up to 30 min ahead.

The corrections applied in our advection-based models to forecast the magnitude of the wind speed observations are necessary due to the tilted trajectories and local effects of the coastline and the cliff. However, the corrections are not perfect. The results are based on a limited amount of dual-setup measurements and it is clear that we could not find a model with a zero mean bias error. The best performing advection model depends on the wind speed and stability. Despite all these limitations, we showed that lidars i) provide range-resolved information to derive site-specific effects influencing the wind speed and ii) are promising candidates for very short-term wind power applications since they can forecast wind speeds with more accuracy than the benchmarks persistence and ARIMA. To use an advection-based wind speed forecasting technique, one could better benefit from horizontal trajectories that do not require height corrections. Additionally, applying this technique in pure offshore areas improves the results, since no corrections due to local effects are required. An operational lidar-based forecasting system on an offshore wind farm would need no corrections at all. Thus, it is reasonable to expect that the forecasting performance of such a system would be better than the best results we have achieved since the many corrections might not have benefited the forecasting accuracy.

Very short-term wind power forecasts typically use statistical techniques that learn from the wind speed and power data at the location of interest and surroundings. Based on our results, a long-range lidar system is likely to decrease the uncertainty in the prediction of offshore wind power, especially during ramp events, i.e., large variation in wind speed within a short period of time, where statistical methods do not perform well.

Our analysis is a first input component to a decision-making model that may include spot market prices, scheduled supply and demand and balancing costs. Thus, here it is not intended to quantify the economic impact of using a lidar-based wind speed forecast. However, as the balancing costs are proportional to the root mean square error, it can be assumed that they will decrease. In particular, as in most of the periods analysed the maximum absolute error is lower than that of the benchmarks,

5 using a lidar-based wind speed forecast might have a positive impact on integrating offshore wind power into the grid.

Further research will focus on using long-range, remotely sensed wind speed observations to predict the power produced by a single wind turbine or a wind farm.

*Author contributions.* Laura Valldecabres conducted the research work and wrote the paper. Alfredo Peña and Michael Courtney extensively contributed to the modelling of the coastal effects and the use of lidar observations for forecasting wind speed respectively. Lueder von

10 Bremen and Martin Kühn supervised the research work and contributed to the structure of the paper. All co-authors participated in the outline and review of the manuscript.

*Competing interests.* The authors declare that they have no conflict of interest.

*Acknowledgements.* This project has received funding from the European Union's Horizon 2020 research and innovation programme under the Marie Sklodowska-Curie grant agreement No 642108. Funding from the ForskEL program to the project "RUNE "No. 12263 and from

15 the Ministry of Science and Culture of Lower Saxony to the project "ventus efficiens "(ZN3024, MWK Hannover) are acknowledged.